# Planning a geostatistical survey to map soil and crop properties: communicating uncertainty and its dependence on sampling density to stakeholders

Christopher Chagumaira[1,2,3,4], Joseph G. Chimungu[4], Patson C. Nalivata[4], Martin R. Broadley[2,3], Alice E. Milne[3], and R. Murray Lark[1,2]

[1]Future Food Beacon of Excellence, University of Nottingham, Sutton Bonington Campus, Loughborough, LE12 5RD, United Kingdom
[2]School of Biosciences, University of Nottingham, Sutton Bonington Campus, Loughborough, LE12 5RD, United Kingdom
[3]Rothamsted Research, Harpenden, AL5 2JQ, United Kingdom
[4]Lilongwe University of Agriculture and Natural Resources, Bunda College, P.O. Box 219, Lilongwe, Malawi

**Correspondence:** Christopher Chagumaira (christopher.chagumaira@nottingham.ac.uk; chris.chagumaira@gmail.com)

**Abstract.**

Much research has examined communication about uncertainty in spatial information to users of that information, but an equally challenging task is enabling those users to understand measures of uncertainty for surveys of different intensity (and so cost) at the planning stage. While statisticians can relate sampling density to measures of uncertainty such as prediction error variance, these do not necessarily help information users (e.g., agronomists, soil scientists, policy makers and health experts) to make rational decisions about how much budget should be assigned to field sampling to produce information of adequate quality. In this study, we considered four ways to communicate uncertainty associated with predictions made based on data from a geostatistical survey, to determine an appropriate sampling density to meet an information user's expectations. These include two methods based on the conditional prediction distribution: the width of prediction intervals, and the joint probability that a particular intervention is required at a random location, but the spatial information indicates the contrary. A third method, the offset correlation is a measure of the consistency of kriging predictions made from data on sample grids with the same spacing but different origins. The implicit loss function is a method which allows the user to reflect on the valuation of losses from decisions based on uncertain information implicit in selecting some arbitrary sampling density. Evaluation of the four communication methods was done through a questionnaire by eliciting opinions of participants with experience in planning surveys, about the method's effectiveness, and comprehensibility of the uncertainty measure and it's trade-off with sampling effort. Our results show significant differences in how the participants responded to the methods, with the joint probability and implicit loss function approaches being not well understood, and offset correlation was the most understood. During feedback sessions, the information users highlighted that they were more familiar with the concept of correlation, with a closed interval of [0,1] and this explains the more consistent responses under this method. The offset correlation will likely be more useful to information users, with little or no statistical background, who are unable to express their requirements of information quality based on other measures of uncertainty.

# 1 Introduction

## 1.1 Mapping to support decisions: importance of spatial mapping

Spatial information is needed to support decisions at different spatial scales. Many approaches can be used to predict soil or crop properties at unsampled locations (e.g. geostatistical methods and machine learning algorithms). These methods make predictions based on a set of point observations configured on a systematic grid or spatial coverage sampling design. Geostatistical methods capture the spatial dependence by modelling the variation as an outcome of a random process (Webster, 2000), whereas machine learning methods do not entail a statistical assumption about the distribution of soil or grain property. Therefore, geostatistical methods offer an approach to sampling because they leverage on the statistical model that provides a basis for planning sampling given a statistical model.

Spatial information is usually derived from field data obtained in surveys. These surveys have costs, for example, travel and logistics, staff costs, time for community engagement, management costs and analytical costs for processing material collected in the field. The denser the sampling the higher the quality of the resulting information (in the sense that the uncertainty attached to spatial predictions is reduced). However, there are diminishing returns to increasing survey effort, and so there is an optimal survey effort where the marginal costs of the survey match the marginal improvement in the resulting information (Lark et al., 2022).

The dependence of the quality of spatial information on survey effort has been studied by geostatisticians. In a geostatistical model the value of a variable at an unsampled location has a prediction distribution, conditional on the model and the data. The variance of this distribution, however, the kriging variance, is conditional on the model only and so can be calculated from the model for any posited set of observations. McBratney et al. (1981) showed how, given a variogram model, ordinary kriging variances could be computed at the cell centres of square sampling grids of different spacing. A plot of kriging variance against spacing could be used to select a sampling grid spacing if a target kriging variance can be specified. A technical challenge is how to obtain the variogram before sampling. One might undertake a reconnaissance survey (particularly when a large final survey is envisaged) to estimate the variogram and use a Bayesian approach to account for its uncertainty (Lark et al., 2017), use a variogram from a cognate area (Alemu et al., 2022), use an average variogram for the variable derived from published studies (Paterson et al., 2018), use a variogram elicited from experts (Truong et al., 2013) or use an adaptive sampling strategy with several phases in which the spatial model is the primary output from early phases (Marchant and Lark, 2006). The general approach of sampling design for ordinary kriging, which McBratney et al. (1981) developed can also be extended to the more general case of spatial prediction from a linear mixed model with spatially correlated random effects and fixed effects which include covariates such as measurements from remote sensors, variables derived from digital terrain models and factorial covariates such as soil maps (Brus and Heuvelink, 2007).

## 1.2 Communicating the uncertainty of spatial information from proposed survey designs

Despite this effort to address the statistical component of survey planning, the generation of measures of uncertainty for particular proposed designs, there has not been a complementary effort on how these measures are understood by information

users, such as survey sponsors, who might have the final responsibility of setting a survey budget, and so determining the quality of the resulting information. Previous studies, including that by Chagumaira et al. (2021), have shown that non-statisticians commonly do not find the kriging variance a meaningful measure of uncertainty to interpret spatial predictions, so it is unlikely they would find it useful as a measure of the quality of survey outputs to balance against costs.

In this study we worked with information user groups (soil science, agronomy, nutrition, and public health) to examine how 60 they interpret measures of survey quality, and whether they regard them as suitable for guiding a decision on the density of samples to be required for a survey. The measures we considered were all ones which could be derived from an initial variogram of the target variable, and we outline them briefly here, more detail is given in Appendix A (A1–A4).

## 1.3 Proposed methods for communicating information quality

We consider two measures derived from the kriging variance as measures of information quality. The first is the prediction 65 interval, the interval which includes the unsampled value with some specified probability. Prediction intervals for surveys on grids of different spacing were proposed, in visual form which allowed the user to evaluate them relative, for example, to differences between critical values of the target variable for management purposes. The second measure was based on the joint probability that a location requires some intervention (because the surveyed variable exceeds or falls below a threshold) and that the spatial prediction at that location indicates the contrary. This was proposed because we found that information users 70 were generally receptive to presentations of uncertain information based on the probability that the mapped variable falls above or below a significant threshold (Chagumaira et al., 2021).

The third measure which we considered is based on value of information theory (Journel, 1984; Lark et al., 2022). It is the implicit loss function (Lark and Knights, 2015). A loss function represents the loss incurred when a decision is based on spatial information which is correct (loss $=0$) or in error (loss $\leq 0$). This is used to analyse quality of information in cases where losses 75 are reasonably straight forward to specify for different scenarios (e.g. Ramsey et al., 2002). Lark and Knights (2015) proposed that, for more complex cases, the implicit loss function might be used in critical assessment of a specified level of survey effort, based for example, on a fixed budget. An implicit loss function is one which, given a model of survey logistics, and statistical information (such as a variogram when the information is obtained by geostatistical prediction) makes a specified survey density the rational choice, i.e. the choice under which a marginal increase in survey cost is equal to the marginal reduction in 80 expected loss when decisions are based on the resulting information. Lark and Knights (2015) proposed that reflection on the implicit loss function would help information users to decide whether a proposed survey budget is consistent with information users' views on the implications of making decisions with uncertain information, and we evaluated that here.

The fourth measure which we considered is the offset correlation. This is a measure of the consistency of spatial information produced when surveying at a particular grid spacing. Lark and Lapworth (2013) considered a hypothetical case in which a 85 variable is mapped by ordinary kriging from data on a sample grid of spacing $\zeta$, a second map is then made of the same variable and from a grid of the same spacing, but in which the origin is shifted from the original grid by $\zeta/2$ in each direction. They showed that, for a specified variogram, the correlation of the mapped values at some location increased as the sampling grid became denser. We suggested that this minimum offset correlation (at a location furthest from a sample point in either grid) is

an intuitive measure of the quality of a survey output, it shows the extent to which the mapped value of the variable is robust
to the location selected as the origin of the survey grid.

Decision-making in the presence of uncertainty is complex, and we have examined it on other related settings Chagumaira
et al. (2022). Our focus in this paper is not the decision-making process (sampling density per se), but on the clarity and
useability of the uncertainty measures for different sampling densities. To do this we presented participants with an explanation
of the method. We provided them with the information according to each method on the relationship between sampling density
and prediction uncertainty for soil pH and grain Se ($Se_{grain}$) concentration. The methods each provide a measure of uncertainty
of the predictions as a function of sampling density. All the measures depend on a common spatial linear mixed model for the
variable, and some also on the location of the marginal distribution. They are therefore mutually consistent, but they do not
provide the same information. Our focus was on accessibility of these methods to information users. The actual decision
process making would be very case-specific, and we would expect that not all of the methods here are relevant in any one case.
We consider this further in the light of our results, in the discussion section.

## 2    Materials and methods

This study was conducted with information users who have been involved with the GeoNutrition project (http://www.geonutrition.
com/), which examined strategies to alleviate micronutrient deficiencies (MND) in Ethiopia and Malawi and included surveys
to provide baseline information on micronutrient concentrations in staple crops and soils, and soil properties (such as pH) which
influence soil to plant transfers of micronutrient. The GeoNutrition project had teams from multiple disciplines (agriculture,
soil science, human nutrition, and public health). It has been shown that concentrations of micronutrients in staple crops and in
soils vary spatially, as do biomarkers for micronutrient status and so interventions to address the deficiencies should be based
on spatial information on all these variables (Gashu et al., 2020; Botoman et al., 2022). The spatial information therefore must
be interpreted by information users from this broad set of disciplines, and all of them might also contribute to decisions on
the amount of effort to be expended on field survey. It is plausible that experts with training in different disciplines might find
different quantitative methods to express uncertainty in information useful for decision-making, and so we recruited a panel
for elicitation which spanned these disciplines.

We recruited the panel from institutions which were partners of the GeoNutrition project research team and the allied
Translating GeoNutrition project in Zimbabwe (ZimGRTA) and the University of Zambia. The participants were volunteers,
recruited from sub-and national-level institutions with responsibility for interventions and policy working on agricultural re-
search and extension services, public health bodies and nutritional research. Soil scientists from the UK were also included.
Panel members were invited by email from the local GeoNutrition/ZimGRTA lead. Most of the participants were familiar
with the GeoNutrition project. They had an interest in contributing to the planning, execution, and interpretation of surveys to
address micronutrient deficiencies, and so were aware of the importance of being able to engage with the process. All these
professionals had experience of delivering advice on research to support implementation of the policy relating to micronutrient
supply and or crop production in their respective countries. However, our focus in on the accessibility of the information on the

effect of sampling intensity and uncertainties, we considered that this experience, from contrasting disciplinary background, is of more importance than the specific context. Ethical approval to conduct this study was granted by the University of Nottingham, School of Biosciences Research Ethics Committees (SBREC202122022FEO) and participants gave informed consent to their participation and subsequent use of their responses. No remuneration was offered, but all participants in African countries who were not able to participate from institutional offices were provided with a one-day data bundle to allow them to join online.

We sought to explore how the information from the sampling conducted in the GeoNutrition project could be used to support decisions about sampling density for other similar projects. We used data from the GeoNutrition project (Gashu et al., 2021), crop and soil properties were measured at national scale in a geostatistical survey conducted in Malawi. In this survey, field sampling was undertaken to support the spatial prediction of micronutrient concentration in crops and soil across Malawi. The sampling design was selected to achieve spatial coverage and used 'main-site' and 'close-pair' sampling to support the estimation of variance parameters of the linear mixed model (Lark and Marchant, 2018). The location of sample points were the centroids of the Delauny polygons, resulting from the stratification function in the spcosa library for the R platform (Walvoort et al., 2010). The sample support (0.1 ha circular plot) for the data consisted of bulk soil and grain samples from aliquots within a single field (Gashu et al., 2020). Therefore, the uncertainty quantification of the predictions relates to the mean values of the target variable across such as support within a field at a specified location, and this is appropriate for deriving regional sampling densities. A detailed description of soil and crop sampling in Malawi are presented by Gashu et al. (2021) and Botoman et al. (2022), and the full data description is provided by Kumssa et al. (2022). Grain and soil samples were prepared and analysed using methods described in Gashu et al. (2020) and Kumssa et al. (2022).

We used variograms for soil pH and $Se_{grain}$ concentration to obtain sampling densities for further notional sampling for an administrative district in Malawi, Rumphi District, with an area of 4769 km$^2$. The outputs were presented to participants in poster format through PowerPoint, and examples of the posters are shown in Figures S5 – S10 in the Supplement.

## 2.1 Format of the exercise

We wanted to elicit from stakeholder the usefulness of proposed methods in helping them assess the implications of uncertainty in spatial prediction in as far as this is controlled by sampling, considering the problem of measuring a soil property and a micronutrient from a crop. Soil pH and $Se_{grain}$ were used as examples for this case study. We invited professionals working in agriculture, nutrition and health at civic organisations, universities, government departments from Ethiopia, Malawi and wider GeoNutrition sites (United Kingdom, Zambia, and Zimbabwe). In total we had 26 participants (18 were agronomists or soil scientists and 8 public health or nutrition specialists).

The elicitation was conducted online using Zoom Video Communications (2022) in two sessions, 26[th] and 28[th] April 2022. There were two sessions in order to accommodate participants from different time zones, and to manage the participants in smaller groups to allow for questions and feedback. The invited participants self-identified as (i) agronomist or soil scientist or (ii) public health or nutrition specialists. The participants also self-assessed their statistical/mathematical background and their frequency of use of statistics in their job role (perpetual, regular and occasional use).

In the exercise, an introductory talk was given to explain the study's objectives. During the talk, we explained the four test methods and how they can be used to assess the implications of uncertainty in spatial predictions to determine appropriate sampling grid space for a geostatistical survey. We explained the structure of the questionnaire to the participants. We emphasized to the participants that we were not testing their mathematical/statistical skills and understanding but rather were testing the accessibility of the methods. We had a feedback session to allow the participants to seek clarification on the presented methods.

The participants considered each method in turn and were asked to select an appropriate sampling grid density based on the method. Evaluation of the test methods was done through a questionnaire, as shown on Table 1. Using the first four questions, Q1 to Q4, we wanted to find out if the method helped the participants to identify a sampling grid spacing. On Q5, we wanted the participants to assess the test methods in terms of their effectiveness in finding an appropriate grid spacing. We asked the participants to rank these methods based on their effectiveness for user in terms of finding a tolerable level of uncertainty in the final survey product, and the comprehensibility of the uncertainty measure and it's trade off with sampling effort. We asked them to put rank 1 as the most effective method and rank 4 the least. The participants recorded their responses using an online questionnaire on Microsoft Forms. The offset correlation was the first method presented to the participants. This was followed by prediction intervals and joint probabilities. The implicit loss function was the final method presented to the participants. We started with a measure we thought all our stakeholders would most easily understand and then moved on to the more complex methods.

## 2.2 Test Methods

### 2.2.1 Statistical modelling and spatial prediction of grain Se concentration and soil pH

We undertook exploratory analysis of soil pH and $Se_{grain}$ concentration using QQ plots, histograms and summary statistics to check whether there was need for transformation of the variables for the assumption of normality. The data for $Se_{grain}$ concentration were skewed and it was necessary to transform them to natural logarithms. The variance parameters for both soil pH and $Se_{grain}$ concentration were estimated by residual maximum likelihood using the likfit procedure in the geoR packages (Diggle and Ribeiro, 2010) for the R platform (R Core Team, 2022) with a constant mean as the only fixed effect. These variance parameters were used in the subsequent test methods. The thresholds we considered, in this study for the prediction intervals and joint probabilities were soil pH of 5 and $Se_{grain}$ concentration of 38 µg kg$^{-1}$. The threshold for soil pH is 5 in Malawi, such that if the pH at a location falls below 5, it would be necessary to apply lime (Chilimba et al., 2013). The threshold $Se_{grain}$ concentration is 38 µg kg$^{-1}$, such that a serving of 330g of grain flour provides a third of the daily estimated average requirement of Se for an adult woman (Chagumaira et al., 2021). The intervention for soil pH was liming, and $Se_{grain}$ was provision of fortified food. Selenium is an essential micronutrient with critical roles in human health, and lack of it can cause thyroid disfunction, and suppressed immune response (Fairweather-Tait et al., 2011).

**Table 1.** The list of questions used to elicit stakeholder opinions about the set of methods that can help end-users to assess the implications of uncertainty in spatial prediction in as far as this is controlled by sampling.

| Number | Question | Responses |
|---|---|---|
| Q1 | We show you here some pairs of example maps of soil pH/$Se_{grain}$, each pair being based on a different grid spacing, and we also show scatter plots which illustrate the strength of the correlation so with a different offset correlation. What do you think is the smallest correlation that would be acceptable if one of the maps were to be used to make decisions? | (1) 0.4<br>(2) 0.5<br>(3) 0.6<br>(4) 0.6<br>(5) 0.8<br>(6) 0.9 |
| Q2 | You are shown different scenarios for the prediction of soil pH/$Se_{grain}$ from different grid spacing's, which determine the width of the prediction interval. What is the grid spacing that gives the widest prediction interval that would be acceptable if one of the maps were to be used to make decisions? | (1) Spacing-20km<br>(2) Spacing-40km<br>(3) Spacing-60km<br>(4) Spacing- 80km<br>(5) Spacing-100km<br>(6) Spacing-120km |
| Q3 | At some location on the map the true value of soil pH/$Se_{grain}$ indicates that an intervention is required, due to error in prediction there is a non-zero probability that the mapped soil pH/$Se_{grain}$ does not show this, this probability increases with grid spacing as shown on the graph. What grid spacing do you think corresponds to the largest acceptable value of this probability? | (1) Spacing-20km<br>(2) Spacing-40km<br>(3) Spacing-60km<br>(4) Spacing-80km<br>(5) Spacing-100km<br>(6) Spacing-120km |
| Q4 | We have three specified implicit loss functions for predictions $Se_{grain}$ concentration over an area of 4,769 square kilometres ($km^2$) for a district/administrative region. With the implicit loss function we assume that the sample density is fixed (e.g. on budgetary grounds) and compute the loss function which would make that a rational choice. We then ask does the loss function implied by the decision look sensible? | (1) Spacing-10km<br>(2) Spacing-20km<br>(3) Spacing-30km |
| Q5 | Please rank these methods in an order of their effectiveness, in your experience, in terms of finding a level of uncertainty that you are able to tolerate when deciding about a sampling grid density. | Rank 1 being the MOST effective and Rank 4 the least |

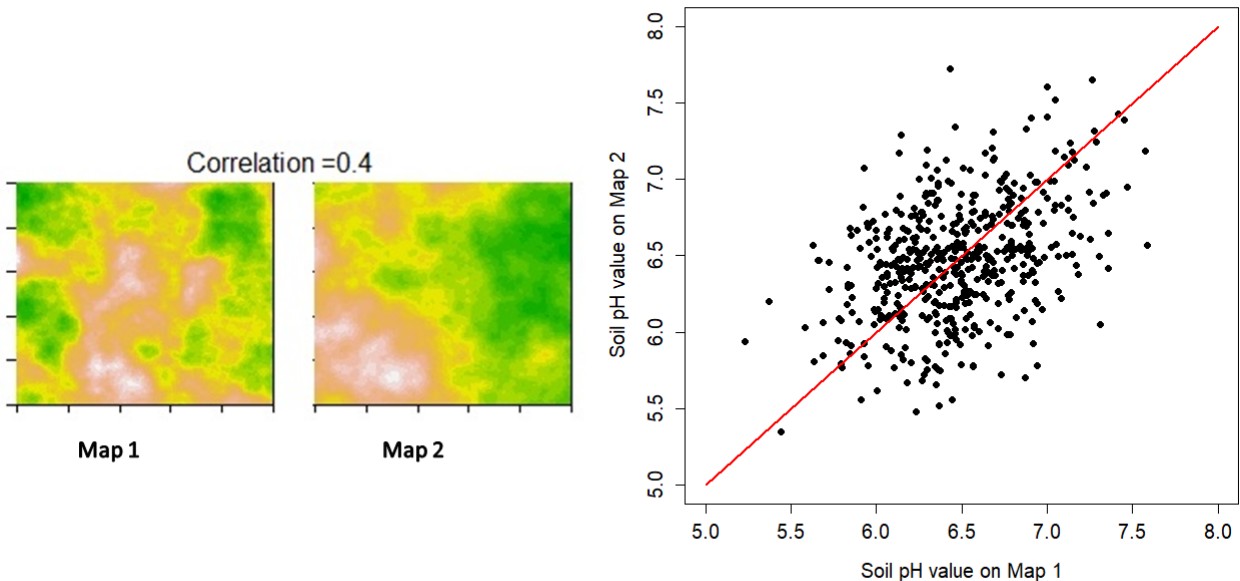

**Figure 1.** The pair of hypothetical maps of pH value and corresponding scatter-plot for offset correlation 0.4.

### 2.2.2 Offset Correlation

We presented the participants with correlated pairs of hypothetical maps of soil pH and $Se_{grain}$, with differing correlations, so that the extent to which maps might differ as a result of the grid offset could be visualized. We also showed scatter plots that illustrated the strength of the correlation. Figure 1, shows an example of pair of maps of soil pH and the corresponding scatterplot (see also Figures S5 and S6). The correlation plots showed the kriging predictions for soil pH or $Se_{grain}$ concentration predicted with parameters estimated in Section 2.2.1. We asked the participants the smallest offset correlation that would be acceptable if one of the maps were to be used to make decisions based on the soil or grain property (see Table 1). Neither of the posited pair of maps, based on offset grids, is to be regarded as closer to reality than the other, the question is how consistent they are.

### 2.2.3 Prediction intervals

Using the variance parameters estimated in Section 2.2.1, we evaluated kriging variances at the centres of cells of square grids of different spacings. We considered minimum and maximum grid spacings of 0.05 and 125 km, respectively, with an increment of 0.5 km. We then computed the cell-centred block kriging variance the spacings we were considering by block kriging (Webster and Oliver, 2007). For all the grid spacings, we computed cell-centred block kriging variance on 0.01 km$^2$

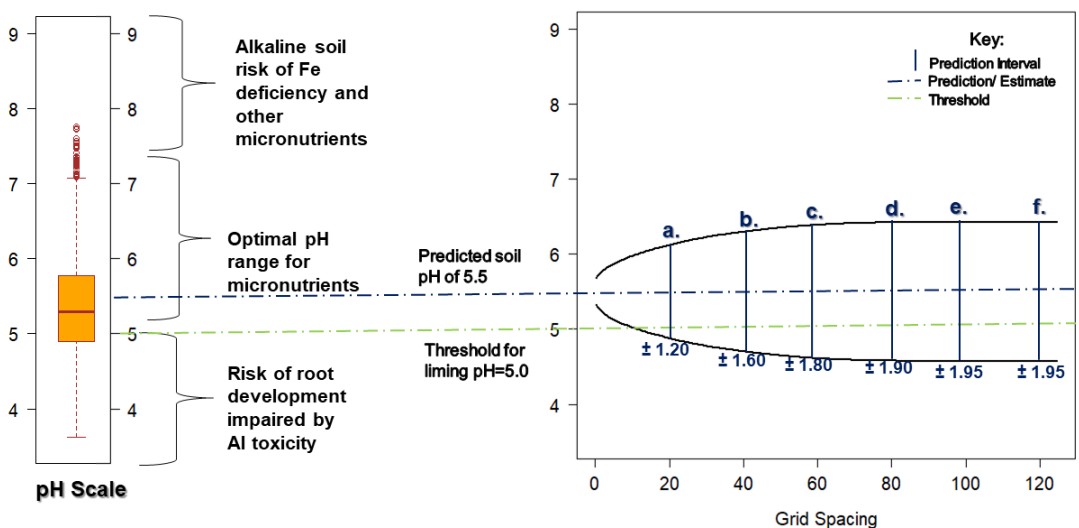

**Figure 2.** An example of a chart, for prediction intervals, with prediction of soil pH of 5.5 with prediction intervals and in relation to a threshold of pH = 5.0.

square blocks. We considered three different predictions for each variable but the prediction interval was fixed, depending only on grid spacing. The three predictions of soil pH were 4.8, 5.5 and 6.0 and those of $Se_{grain}$ were 20, 55 and 90 μg kg$^{-1}$. The predictions of soil pH and $Se_{grain}$ concentration were presented to the participants in a chart as shown in Figure 2.

The chart consisted of (a) box plot of the distribution of the measured variable based on all soil samples from the study area, (b) a graph of the lower and upper prediction intervals for the prediction at the point of interest for grid spacings from 0 to 120 km, and lines indicating (c) the $z_t$ and (d) the prediction (see Figures 2, S7 and S8). From the chart, we asked the participants to select the grid spacing that gives the widest prediction interval that would be acceptable if the mapped predictions were to be used to make decisions about soil management or interventions to address human Se deficiency (see Table 1).

### 2.2.4 Joint probability

The joint probability is a measure of uncertainty in terms of the risk of failing to intervene at some location given that an intervention is needed. We presented the participants with a chart of joint probabilities plotted against grid spacing is shown on Figure 3 (see also Figure S9), and this probability increases with grid spacing. The joint probability is bounded on an interval [0,1]. A probability of 1, indicates that the prediction will be equivalent to the overall mean of the dataset. If the prediction of $Se_{grain}$ or pH was below the threshold, $z_t$, an intervention is needed. We then asked the participant what grid spacing they thought corresponded to the largest acceptable value of this probability (see Table 1).

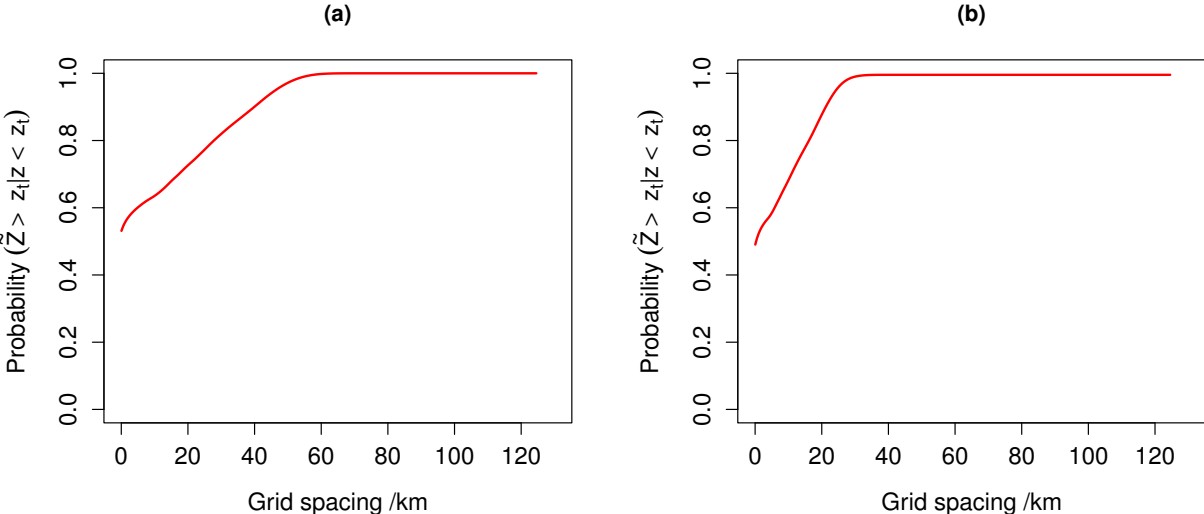

**Figure 3.** An example of the chart of joint probabilities plotted against grid spacing for (a) soil pH and (b) $Se_{grain}$ concentration. At a location $\mathbf{x}_0$, $\tilde{Z}$ is the prediction and $z$ is the value of the variable at that location.

### 2.2.5 Implicit loss functions

In order to compute the implicit loss function, we needed a cost model for Rumphi district. We used the function defined in Lark and Knights (2015) to return the costs of $n$ samples over an area $A$ km$^2$, i.e. a sample density of $r = N/A$ samples per km$^2$ :

$$C(n) = \omega + vAr + \beta At_r, \tag{1}$$

where $\omega$ are the fixed costs, $v$ cost of laboratory analysis per unit, and $\beta$ the field costs per work day per team. The quantity $t_r$ is time taken to sample per km$^2$ at a density of $r$ per km$^2$. We obtained these costs for Rumphi district by considering the available costs for crop sampling during the GeoNutrition survey conducted in Malawi at national-scale (Gashu et al., 2021; Kumssa et al., 2022). A detailed description of how the costs were computed is presented in the Supplementary Material.

We fixed the asymmetry ratio at 1.5, assuming the elicited mean probability threshold from similar stakeholders in Ethiopia and Malawi (Chagumaira et al., 2022) can be regarded as an approximation of $P_0$ which corresponds to a quantile of prediction distribution. This implied a bigger loss for overestimation of the variables (i.e. failing to intervene of $Se_{grain}$ are smaller than prediction). With the implicit loss function we assumed that the sample density is fixed (e.g. on budgetary grounds) and computed the loss function which would make that a rational choice. We presented three specified implicit loss functions for predictions of $Se_{grain}$ for Rumphi district, with an area of 4,769 km$^2$ with sampling densities fixed at 10, 20 and 40 km. Figure 4 (see also Figure S10), shows the implicit loss function for $Se_{grain}$. We then asked the participants to identify the loss

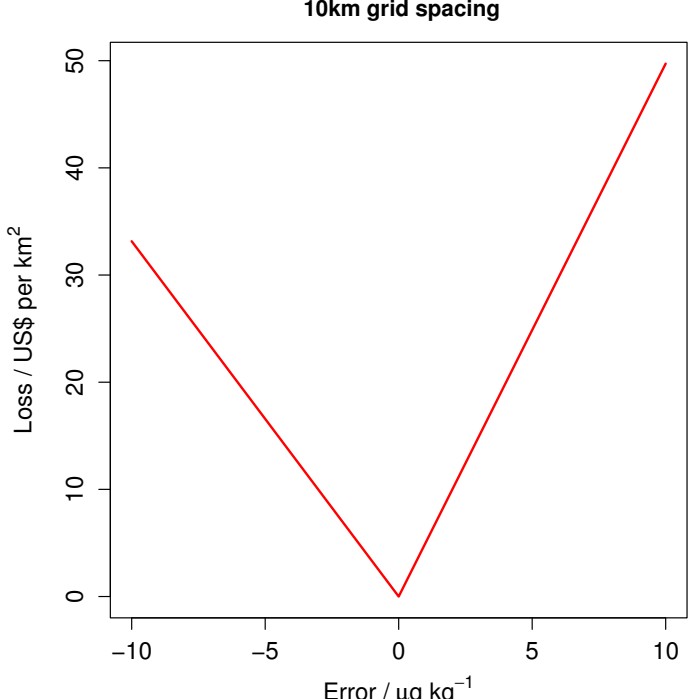

**Figure 4.** An example of specified implicit loss functions for predictions of $Se_{grain}$ concentration at a 10km grid spacing.

function implied by the sampling decision that looked more plausible to make decisions about interventions to address human Se deficiency (see Table 1).

## 2.3 Data Analysis

### 2.3.1 Test methods

The responses for Q1 to Q4 were presented as contingency tables. The contingency table allowed us to evaluate whether there were differences in the responses of the participants based on (i) variable used in the test method, (ii) professional group and (iii) by frequency of use of statistics. We analysed the contingency table on the basis of a null hypothesis that the distribution of observations between responses (e.g. selected grid spacing) is independent of the factor in the column (e.g. professional group). If evidence is provided to reject the null hypothesis, then this would indicate that how a respondent interprets the information

presented to select a grid spacing depends on their professional group. A detailed description of how we used the contingency tables to partition the responses of the questionnaire are presented in Appendix B.

    The rows of each table correspond to the response (e.g. the different grid spacings) and, the full table, the columns correspond to the frequency of use of statistics, nested, within professional group and nested within variable used (soil pH or $Se_{grain}$).

Contingency tables allowed us to test the null hypothesis of random association of responses with the different factors in the columns. The expected number of responses under the null hypothesis, $e_{i,j}$ in a cell $[i,j]$, is a product of row ($n_i$) and column ($n_j$) totals dived by the total number of responses ($N$), and this the null hypothesis of the contingency table which is equivalent to an additive log-linear model of the table. An alternative to the additive model for the contingency table, is the saturated model that has an extra $(n_r - 1)(n_c - 1)$ term that allows for interaction amongst the columns and tables of the table. The proportions of observed responses $o_{i,j}$ may differ from $e_{i,j}$ in a cell $[i,j]$ and the likelihood ratio statistic or deviance, $L$, can be used to provide evidence against the null hypothesis. The likelihood ratio statistic is computed by

$$L = 2 \sum_{i=1} \sum_{j=1} o_{i,j} \log \frac{o_{i,j}}{e_{i,j}}. \tag{2}$$

where $L$ has an approximate $\chi^2$ distribution under the null hypothesis of random association between the rows and columns of the table, with $(n_r - 1)(n_c - 1)$ degrees of freedom (Christensen, 1996; Lawal, 2014). We fitted the log-linear models using the loglm function from the MASS package (Venables and Ripley, 2002) for the R platform.

In our study, we wanted to find out if the responses recorded by the stakeholders depended on the variable used (soil pH or $Se_{grain}$ concentration), and background of the respondent. We expected the responses to differ. We thought the stakeholders would have different perceptions of the impacts of the uncertainty for soil pH and $Se_{grain}$ concentration. There were more agronomist or soil scientists than public health or nutrition specialists in the meeting, and we expected the priorities of the groups to differ when making interventions for soil pH and $Se_{grain}$ concentration. We also thought the frequency of use of statistics would influence the choice of method used to select an appropriate grid spacing.

We first tested for differences in responses recorded for each test method, by the variable used (soil pH or $Se_{grain}$ concentration) using contingency tables. The responses from stakeholders in different professional groups were pooled within the two variables, as illustrated by the Pooled table 1 on Table 2. This gave us a six (responses) by two (variables) contingency table with 5 degrees of freedom for the questions corresponding to offset correlation, prediction intervals and joint probabilities (Q1 to Q3). However, for the implicit loss function we did not consider this because we only had a loss function for $Se_{grain}$ concentration.

Second, we considered if the differences in the responses depended on the professional group of the respondent. Finally, we considered whether the frequency of use of statistics in their job role had an impact on the responses recorded by the respondents. For some questions, we noted difference in the responses when pooled within variable used (soil pH or $Se_{grain}$ concentration) and there was no differences in responses in professional groups and frequency of use of statistics for all questions. We further analysed the pooled tables or separate subtables to examine if the responses were uniformly distributed and the null hypothesis is a random distribution. We wanted to test whether the responses of the participants were uniform, i.e., each grid spacing has equal likelihood of occurrence.

### 2.3.2 Assessment of the method

The responses for the Q5 were tabulated with the methods as the columns and ranks as the rows. The participants ranked their preferred method first. However, to calculate the mean rank, $\bar{r}_i$, for each method for all the respondents, we assigned a score of

4 for the most preferred method and 1 for the least. We computed the $\bar{r}_i$ for each method for all respondents. We then separated the respondents by their professional group and computed the mean ranks.

Finally we separated the respondents by their frequency of use of statistics in their job role. Under a null hypothesis of random ranking for set of $k$ ranks, the expected mean rank is $(k+1)/2$. The evidence against this hypothesis is measured a statistic distributed as $\chi^2(k-1)$:

$$\frac{12n}{k(k+1)} \sum_{i=1}^{k} \left\{ \bar{r}_i - \frac{k+1}{2} \right\}, \tag{3}$$

where $n$ is the total number of rankings (Marden, 1996).

## 3   Results

There was reasonably even spread in terms of the location of our participants, see Figure B1 (Appendix B). About 54% of the participants were constantly using statistics/mathematics within their job role. Only a few participants were educated to the level of certificate/diploma (8%).

### 3.1   Test methods

#### 3.1.1   Offset correlation

The full table for Q1 and the subsequent subtables are presented in the Appendix C (Tables C2–C4). There were no differences in the responses when the columns were pooled by the variable used, soil pH vs. $Se_{grain}$ concentration, $p = 0.656$ (Table 2). There were no differences in the responses when the columns were also pooled within professional groups ($p = 0.491$) and frequency of use of statistics ($p = 0.595$). Further analysis of the question on offset correlation was based on pooled counts, see Table C4. There was strong evidence to reject the null hypothesis that the responses are uniformly distributed ($p = 0.003$). Figure 5 shows the responses of how all the participants responded to Q1, for offset correlation. Most of the respondents selected offset correlation of $0.7$ as the smallest offset correlation that would be acceptable if one of the maps were to be used to make decisions based on the soil or grain property. We extracted the grid spacings, for soil pH and $Se_{grain}$, corresponding to the selected offset correlation of $0.7$. The spacings were extracted from a plot of offset correlation against grid spacing obtained from the variance parameters of each variable (see Figure S4). The grid spacing for soil pH is 25 km and for $Se_{grain}$ is 12.5 km. The grid spacing corresponding to the offset correlation for each variable were computed from the variogram of the variable.

#### 3.1.2   Prediction interval

There were no differences in the responses when pooled within the variable used, $p = 0.656$, for prediction intervals, see Table 3. We then pooled the responses within the professional groups, and there was no evidence to reject the null hypothesis ($p = 0.498$). Also, there were differences when responses were pooled within frequency of use of statistics, $p = 0.152$. There-

**Table 2.** Analysis of the question on offset correlation, Q1, according to variable used, professional group and frequency of use of statistics.

| | Deviance ($L^2$) | Degrees of freedom | $P$ |
|---|---|---|---|
| **Full contingency table analysis** | | | |
| Full table | 54.57 | 55 | 0.491 |
| Pooled by variable used (pH v. $Se_{grain}$) | 3.29 | 5 | 0.656 |
| Pooled by professional group | 6.50 | 5 | 0.260 |
| Pooled by frequency of use of statistics | 8.35 | 10 | 0.595 |
| **Subtable–pooled counts: variable used** | | | |
| Soil pH | 27.01 | 25 | 0.352 |
| $Se_{grain}$ | 24.2 | 25 | 0.507 |
| **Subtable–pooled counts: professional group** | | | |
| Agronomist or soil scientist | 26.25 | 25 | 0.394 |
| Public health or nutrition specialist | 21.81 | 25 | 0.646 |
| **Subtable–pooled counts: frequency of use of statistics** | | | |
| Perpetual use of statistics | 8.99 | 15 | 0.878 |
| Occasional use of statistics | 18.17 | 15 | 0.254 |
| Regular use of statistics | 19.06 | 15 | 0.211 |
| **Subtable–pooled counts** | | | |
| Responses are uniformly distributed | 17.69 | 5 | 0.003 |

fore, further analysis of the question on prediction intervals was based on pooled counts of responses. There was no evidence to reject the null hypothesis that the responses are uniformly distributed ($p = 0.169$). Figure 6 shows the bar charts of how all the participants responded to the Q2 for prediction intervals. For this method, there no clear choice of grid spacing for sampling for soil pH and $Se_{grain}$.

### 3.1.3 Joint probabilities

Table 4 shows the results for partitioning the contingency table for the question on joint probabilities, Q3. There was strong evidence to reject the null hypothesis when the columns were pooled by variable used, $p \leq 0.001$. Therefore, further analysis was based on separate subtables for soil pH and $Se_{grain}$ concentration. For both variables, there were no differences in the responses when the columns were pooled within professional groups and frequency of use of statistics. Both soil pH and $Se_{grain}$ concentration, there was strong evidence to reject the null hypothesis that the responses were uniformly distributed ($p \leq 0.001$). The bar charts for the responses for the question on joint probabilities for soil pH are presented in Figure 7a, and

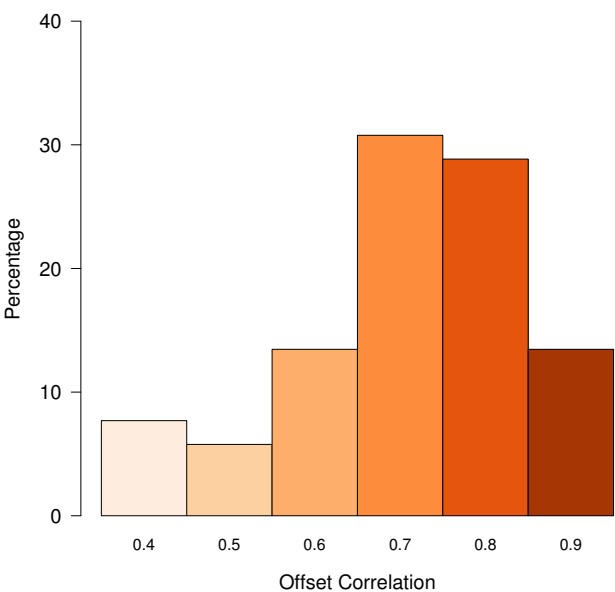

**Figure 5.** Bar charts showing how the participants responded to Q1 for offset correlation.

the grid spacing was 60 km. The responses for $Se_{grain}$ concentration are presented in Figure 7b, and the grid spacing was 40 km.

### 3.1.4 Implicit loss functions

The results for partitioning the contingency table for implicit loss function, Q4, are presented in Table 8. There were no differences in the responses when the columns of the table were pooled within professional groups ($p = 0.781$) and frequency of use of statistics ($p = 0.828$). Further analysis of the question on implicit loss function was based on pooled counts of responses. There was strong evidence to reject the null hypothesis that the responses were uniformly distributed ($p \leq 0.001$). The bar charts for the responses pooled counts for all respondents are shown on Figure 8. The grid spacing chosen by the 325    participants for $Se_{grain}$ concentration is 20 km.

### 3.2   Assessment of the test methods

The question on ranking of the method was analysed in three ways. At first we computed the mean ranks for all participants and tested for the evidence against the null hypothesis of random ranking. There was strong evidence to reject the null hypothesis of random ranking, $p \leq 0.001$ (Table 6). Then the mean ranks were computed for each professional group and there was strong 330    evidence to reject the null hypothesis of random ranking in each group ($p \leq 0.001$). Finally, we separated the participants into

**Table 3.** Analysis of the question on prediction interval, Q2, according to variable used, professional group and frequency of use of statistics.

|  | Deviance ($L^2$) | Degrees of freedom | $P$ |
|---|---|---|---|
| Full contingency table analysis | | | |
| Full table | 56.0 | 55 | 0.437 |
| Pooled by variable used (pH v. Se$_{\text{grain}}$) | 0.972 | 5 | 0.965 |
| Pooled by professional group | 4.36 | 5 | 0.498 |
| Pooled by frequency of use of statistics | 14.5 | 10 | 0.152 |
| Subtable–pooled counts: variable used | | | |
| Soil pH | 23.8 | 25 | 0.531 |
| Se$_{\text{grain}}$ | 31.2 | 25 | 0.181 |
| Subtable–pooled counts: professional group | | | |
| Agronomist or soil scientist | 26.5 | 25 | 0.381 |
| Public health or nutrition specialist | 25.1 | 25 | 0.455 |
| Subtable- pooled counts: frequency of use of statistics | | | |
| Perpetual use of statistics | 9.68 | 15 | 0.840 |
| Occasional use of statistics | 16.88 | 15 | 0.330 |
| Regular use of statistics | 15.08 | 15 | 0.450 |
| Subtable- pooled counts | | | |
| Responses are uniformly distributed | 7.77 | 5 | 0.169 |

three groups according to their frequency of use of statistics in the job role, and computed the mean ranks in each group. There was strong evidence to reject the null hypothesis of random ranking in each group ($p \leq 0.001$).

Amongst all the respondents, the offset correlation was ranked as the most effective (Figure 9a) and implicit loss function as the least effective. Both professional groups (i.e. agronomist or soil scientist and public health or nutritionist) ranked offset
correlation first but differed in the second and least ranked methods (Figure 9b to 9c). Public health or nutrition specialists ranked second prediction intervals and implicit loss function as the least effective. The agronomist or soil scientist group ranked prediction intervals as the least effective and joint probabilities as second.

When respondents were separated by their frequency of use of statistics, offset correlation was also ranked first (Figure 9d to 9f). Those who use statistics occasionally, in their job role, ranked the implicit loss function as the second best
and the prediction intervals the least. Joint probabilities were ranked second and implicit loss function as the least effective by those who regularly use statistics in the job role. Those who always use statistics, ranked conditional probabilities second. Prediction intervals and implicit loss functions were ranked last.

**Table 4.** Analysis of the question on joint probabilities, Q3, according to variable used, professional group and frequency of use of statistics.

| | Deviance ($L^2$) | Degrees of freedom | $P$ |
|---|---|---|---|
| Full contingency table analysis | | | |
| Full table | 60.6 | 55 | 0.281 |
| Pooled by variable used (pH v. Se$_{grain}$) | 26.7 | 5 | $< 0.001$ |
| Pooled by professional group | 5.32 | 5 | 0.378 |
| Pooled by frequency of use of statistics | 14.5 | 10 | 0.152 |
| Subtable–pooled counts: variable used | | | |
| Soil pH | 12.1 | 25 | 0.986 |
| Se$_{grain}$ | 21.8 | 25 | 0.647 |
| Soil pH subtable–pooled counts: professional group | | | |
| Pooled within professional group | 4.48 | 5 | 0.483 |
| Agronomist or soil scientist | 3.10 | 10 | 0.979 |
| Public health or nutrition specialist | 4.50 | 10 | 0.922 |
| Soil pH subtable–pooled counts: frequency of use of statistics | | | |
| Pooled within frequency of use of statistics | 0.889 | 10 | 1.00 |
| Perpetual use of statistics | 4.50 | 5 | 0.480 |
| Occasional use of statistics | 4.36 | 5 | 0.499 |
| Regular use of statistics | 2.33 | 5 | 0.802 |
| Soil pH subtable–pooled counts | | | |
| Responses are uniformly distributed | 50.15 | 5 | $< 0.001$ |
| Se$_{grain}$ subtable–pooled counts: professional group | | | |
| Pooled within professional group | 4.77 | 5 | 0.445 |
| Agronomist or soil scientist | 11.0 | 10 | 0.361 |
| Public health or nutrition specialist | 6.09 | 10 | 0.808 |
| Se$_{grain}$ subtable–pooled counts: frequency of use of statistics | | | |
| Pooled within frequency of use of statistics | 9.55 | 10 | 0.481 |
| Perpetual use of statistics | 1.73 | 5 | 0.886 |
| Occasional use of statistics | 5.55 | 5 | 0.353 |
| Regular use of statistics | 4.99 | 5 | 0.417 |
| Se$_{grain}$ subtable–pooled counts | | | |
| Responses are uniformly distributed | 36.77 | 5 | $< 0.001$ |

**Table 5.** Analysis of the question on implicit loss function, Q4, according to variable used, professional group and frequency of use of statistics.

|  | Deviance ($L^2$) | Degrees of freedom | $P$ |
|---|---|---|---|
| Full contingency table analysis |  |  |  |
| Full table | 8.91 | 10 | 0.541 |
| Pooled by professional group | 0.49 | 2 | 0.781 |
| Pooled by frequency of use of statistics | 1.49 | 4 | 0.828 |
| Subtable–pooled counts: professional group |  |  |  |
| Agronomist or soil scientist | 2.33 | 4 | 0.676 |
| Public health or nutrition specialist | 6.09 | 4 | 0.193 |
| Subtable- pooled counts: frequency of use of statistics |  |  |  |
| Perpetual use of statistics | 1.73 | 2 | 0.422 |
| Occasional use of statistics | 1.73 | 2 | 0.422 |
| Regular use of statistics | 3.96 | 2 | 0.138 |
| Subtable- pooled counts |  |  |  |
| Responses are uniformly distributed | 54.00 | 2 | $< 0.001$ |

**Table 6.** Analysis of Q5 according to professional group and level of use of statistics in job role

|  | Test Statistic ($X^2$) | Degrees of freedom | $P^*$ |
|---|---|---|---|
| All respondents | 61.1 | 3 | $< 0.001$ |
| Professional group |  |  |  |
| Agronomist or soil scientist | 49 | 3 | $< 0.001$ |
| Public health or nutrition specialist | 15.6 | 3 | $< 0.001$ |
| Frequency of use of statistics |  |  |  |
| Perpetual user of statistics | 34 | 3 | $< 0.001$ |
| Occasional user of statistics | 28.5 | 3 | $< 0.001$ |
| Regular user of statistics | 49.8 | 3 | $< 0.001$ |

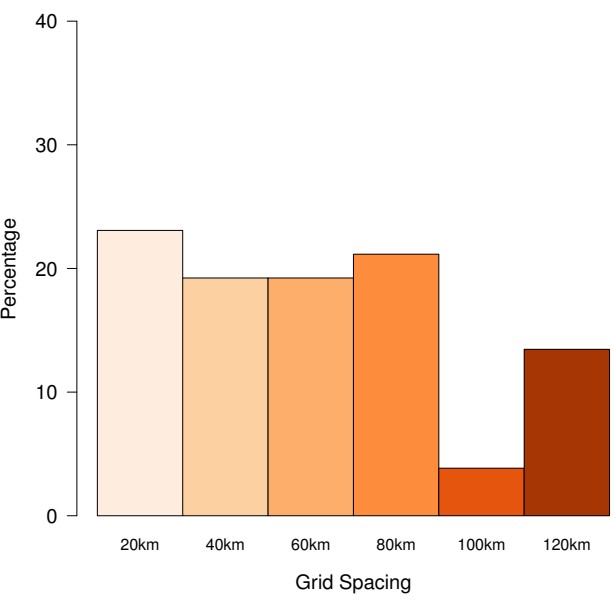

**Figure 6.** Bar charts showing how the participants responded to the Q2 for prediction intervals.

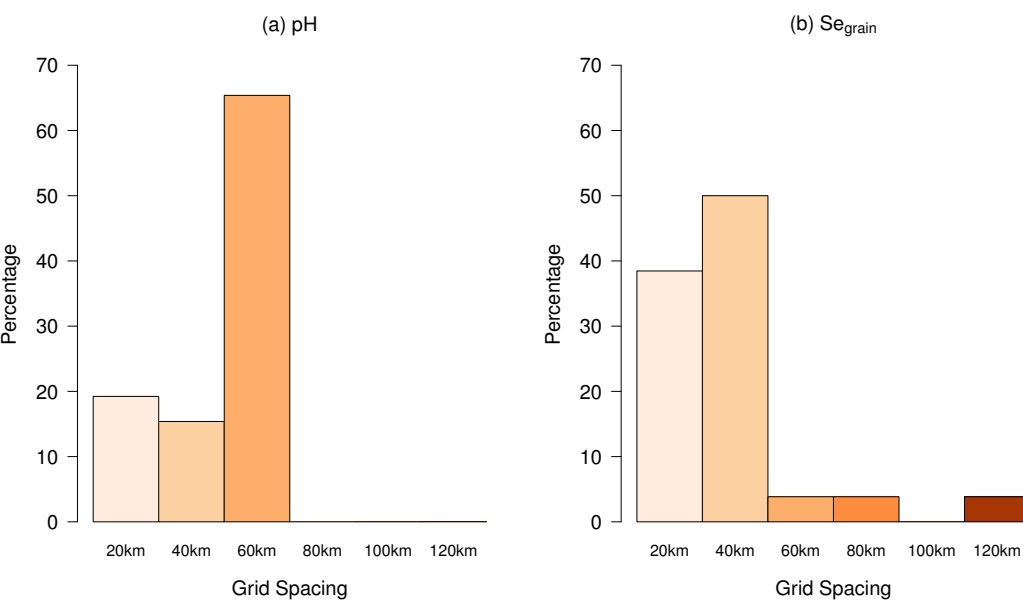

**Figure 7.** Bar charts showing how all the participants responded to the Q3 for joint probabilities for (a) soil pH and (b) $Se_{grain}$ concentration.

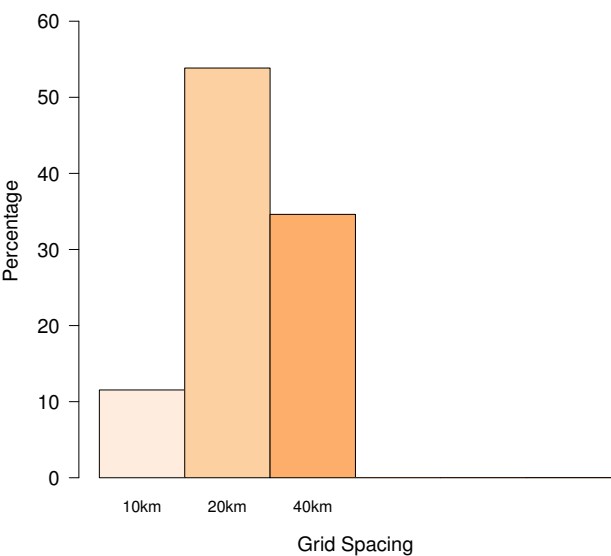

**Figure 8.** Bar charts showing how all the participants responded to the Q4 for implicit loss function.

## 4    Discussion

In this study, we presented to groups of information users, four methods (offset correlation, prediction intervals, joint prob-
abilities and implicit loss functions) that can be used to support decisions on sampling grid spacing for a survey of soil pH
and $Se_{grain}$. We wanted to find out if the information users had a preference among the approaches presented to them. Offset
correlation was ranked first as the method the stakeholders found easy to interpret (see Figure 9) in order to a decision on
sampling density. Most respondents (30 %) selected an offset correlation of 0.7, and slightly fewer selected 0.8 so over half
of respondents are accommodated within this range of values. During the feedback session, information users highlighted that
they were more familiar with the concept of correlation, with a closed interval of [0,1]. This is likely to explain the consistency
of the results for this criterion, with over half the respondents selection 0.7 or 0.8 as a minimum acceptable correlation. Our
results are consistent with findings of Hsee (1998), that relative measures of some uncertain quantity (Hsee gives an example
of the size of a food serving relative to its container) are more readily evaluated than absolute measures (the size of serving).
An easy-to-evaluate attribute, such as the bounded correlation of [0,1], has a greater impact on a person's judgement of utility.
Hsee (1998) describe this as the "relation-to-reference" attribute. It is therefore, not surprising that the offset correlation is
highly-ranked.

The offset correlation seems to be a criterion which respondents are more likely to find comprehensible, and so a basis for
selecting the sample density for a geostatistical survey, than alternatives such as kriging variance. Furthermore, it appears to be

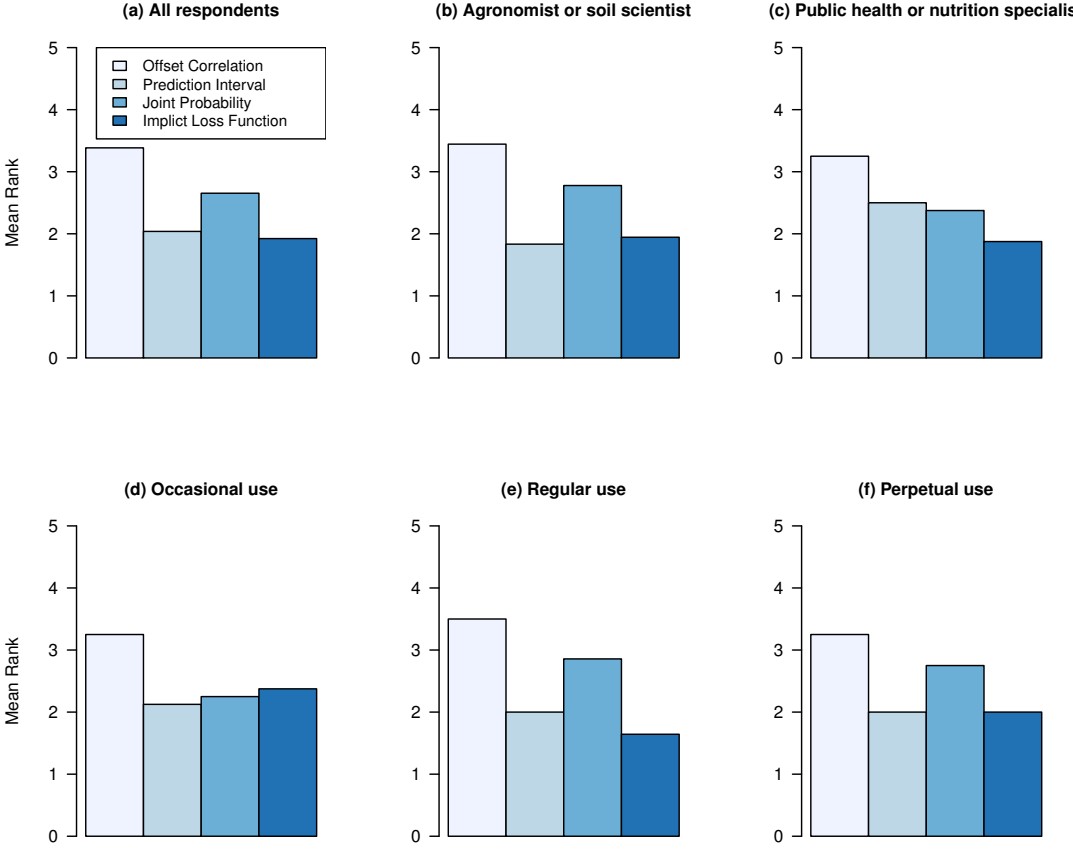

**Figure 9.** Ranking of test methods in terms on the most effective: (a) by all respondents, professional group: (b) agronomists or soil scientist and (c) public health or nutritionist specialists, and frequency of use of statistics: (d) occasional use, (e) regular use and (f) perpetual use.

a measure of uncertainty which participants in the study found comprehensible, and so were able to use to select a grid sample
spacing. It recognises that spatial variation means that maps interpolated from offset grids will differ but that the more robust
the sampling strategy the more consistent they will be. However, Chagumaira et al. (2021) found that measures of uncertainty
related to a specific management threshold of the mapped variable were preferred by participants for the interpretation of
uncertain spatial information to general quality measures without a specific management or policy implication. In this case,
in contrast, the preferred criterion, the offset correlation, is a general measure of map quality, which is not directly linked to
specific interpretation.

Joint probabilities were ranked second. Under this method, the information users selected spacings where joint probabilities
was 1.0 or very close, i.e. the prediction equivalent to the overall mean. This suggests that the information users may not have
fully understood the method. This finding is consistent with the general view that users of information commonly find prob-

abilities difficult to interpret (Spiegelhalter et al., 2011). Because probabilities are bounded [0,1], the 'relation-to-reference" attribute effect by Hsee (1998) may explain the previous preference for joint probabilities (Jenkins et al., 2019; Chagumaira et al., 2021), but information users still struggle to interpret them correctly. Perhaps if the problem had been framed in a different way, the information users may have understood this method much better. More work is needed to investigate if framing the joint probabilities in a different way would improve the judgement of utility of the information users. More examples and more illustration may be needed in order to 'prime' the participants before the exercise. A method might be regarded as easy to interpret, because of its form, even when it is not (in this case a large value of the probability indicated that there was no spatial information in the map to make its predictions better than the overall mean).

Prediction intervals were ranked third by all the respondents, but there was no evidence against the null hypothesis of random selection among the available spacings. During a feedback session, the information users cited difficulties of assessing the significance of a given prediction interval given that it can be associated with different prediction values. For very large or small prediction values the uncertainty is immaterial, it is near decision threshold that it becomes important. Similarly, prediction intervals were not highly ranked by information users for communicating uncertainty in maps (Chagumaira et al., 2021). Similar reasons were given by the respondents. We expected that prediction intervals would be of greatest value for specific interpretation of particular sites but would be of limited value for survey planning.

The implicit loss functions was the lowest-ranked method. The group also commented that they had difficulties understanding this method, and most people opted for the central value. Loss functions are not readily accessible. It is difficult to define a loss function because it requires the cost of the errors, and we tried to show information users some consistent approach with some plausible design. The fact that they did not understand the loss functions, shows there is need for more specific examples to help information users think about loss function and their implications. It might help the information users to provide some quantitative information about the costs of the survey, cost associated with intervention campaigns and costs of the impacts on MND on a country's gross domestic production. A reflection of these would allow the information user to use these implicit assumptions when they were making decisions for selecting a fixed grid spacing for working with (Lark and Knights, 2015).

The background of the information users, i.e., professional group and frequency of use of statistics, had no influence on their responses for all the methods. However, the background of the information users had an influence on their ranking of the methods in terms of their effectiveness. The offset correlation was ranked as the most effective by all professional groups and by all respondents separated by frequency of use of statistics. Prediction intervals were ranked least effective by those respondents who identified as agronomist or soil scientist, but were ranked second by those in public health or nutrition.

At the beginning of the online workshop, we explained each method with the aid of illustrations. After an explanation of each method, there was a feedback session to allow the participants opportunities to seek clarity on ambiguous and unfamiliar concepts from the presenters. The participants' questions were answered and explained in different ways by CC, RML and AEM, with the use of illustrations. However, there are limitations with online workshops. Most participants would have the cameras switched off, and the "unconscious" feedback to presenters by observing the reactions of participants could not be noticed as during in-person workshops. The "unconscious" feedback would prompt the presenter to use a different approach to explain unfamiliar concepts and ambiguous terms. Due to internet connectivity, online workshops are timed and there will less

time for feedback sessions. In such instances, respondents may seek clarity from the colleagues who have the same interests, resulting in bias (Ball, 2019).

All the methods may give different results for different variables, because they depend on the variogram of the variable in question. There maybe different grid spacings selected for the different variables. A potential problem may exist, if the variables were to be sampled in one survey and what spacing should be used? This is an important question that needs to be addressed when planning for soil and crop sampling. It may be reasonable to opt for the grid spacing for the variable that maybe the hardest to characterise. Another option would to consider some minimum quantile over all variables through a group elicitation. Black et al. (2008) proposed that a critical subset of soil properties are identified such that the overall sampling scheme is satisfactory for all of the so-called 'canary indicators'.

All the information users recruited in this study were employed in public sector institutes (e.g., universities, civil organisations, research, and extension) and had experience in their respective fields in an SSA setting. We had no basis for a power analysis to identify a sample size for this activity. Given the exploratory nature of this research, our primary aim was to capture insights from as many relevant participants as possible within each institution. As a result, our major consideration was recruiting individuals willing to participate and with experience in their respective institutions. We therefore attempted to recruit the entire set of suitable respondents in each country. We recognize that the small sample size limits the generalizability of statistical findings. While this study provides insights into participant perspectives by specialism, the lack of demographic information—such as gender, age, location, and years of experience—limits the depth of analysis. These characteristics may impact responses; for example, different age groups or experience levels might prioritize certain issues differently. Future studies should consider including these demographic details to explore how such factors influence perspectives, thus enhancing the robustness of the findings and allowing for subgroup analysis. For this reason, we have interpreted results cautiously and have also incorporated qualitative insights from participants to provide a richer context for understanding these early findings. Moving forward, we plan to include an initial power analysis and possibly extend the study through broader collaborations to enhance robustness.

As we have emphasized, in our objectives, this study was to examine the accessibility of different measures of uncertainty to professional stakeholders. While all methods depend on a common linear mixed model for all the variables, they provide differing, although consistent, information. The different measures would each be most useful for sampling planning for geostatistical prediction to address particular problems. We illustrate this in Fig 10. In scenario I, is a general geochemical survey prior to a detailed MND study to show likely spatial variation of deficiency and identify location for detailed experimentation. In this case there are no specific decisions to be made using the map and there are no costs (e.g. costs of interventions) linked to the outcomes under considerations. So the implicit loss functions and joint probabilities (there are no thresholds) will not be useful. However, the prediction intervals and offset correlation will be very useful. Due to the lack of specificity in use of the maps and associated costs the decision will be relatively straightforward for the range of stakeholders in consideration. Scenario II is a detailed geochemical survey after the exploratory MND study with specific objectives to identify uncertainties and greatest risk over a sub-region, and plan intervention strategies to address deficiencies. Here there are quantifiable losses under the errors and the information will be required information important policy-implemented tasks. Given these consider-

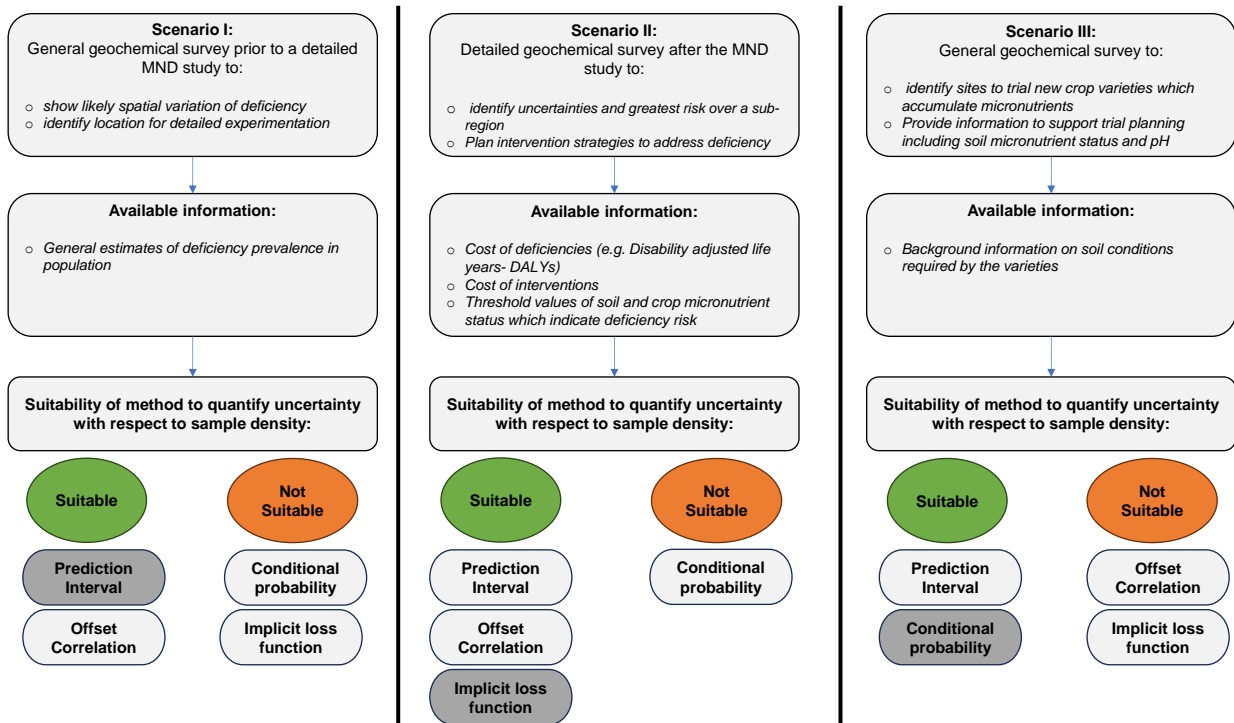

**Figure 10.** Illustration of the different scenarios of geochemical surveys and suitability of the method to quantify uncertainty with respect to sampling densities.

ations, the offset correlation will be too general and therefore not very useful. The threshold for making decision to intervene

is known and therefore the joint probabilities will be useable and useful. However, there will be need to invest in specific

training for specialists (e.g. public health and nutrition experts), and to elicit a probability threshold that represents a consensus

model of the losses associated with making a decision with uncertain data. This could follow Chagumaira et al. (2022), who

elicited a mean probability threshold for making decisions for making decisions about interventions to address Se deficiency.

The implicit loss function will also be useful in this scenario and can be used to judge resource allocation between this and

445 general food fortification. Scenario III is general geochemical survey to identify sites to trial new crop varieties that accumu-

late micronutrients, and to provide information to support trial planning such as soil micronutrient status and pH. Background

information on soil conditions required by the varieties will be available. In this case, the offset correlation is too general a

measure to be useful, and implicit loss functions are not relevant. However, the joint probabilities will be very relevant, for

example, once could compute the probability that a valuable potential site is not identified. The prediction interval will also be

useful in this case, but not as direct as the joint probabilities. There will be need for training give the challenges our participants

faced in understanding the method– e.g, use of games and visual examples of everyday problems to build confidence in the

interpretation of probabilities to support decision making.

## 5 Conclusions

In this exploratory study we evaluated four methods of communicating uncertainty associated with kriging predictions made from data from a geostatistical survey, to determine an appropriate sampling density to meet information users expectations. Users of information on soil variation need accessible ways of understanding the implications of sampling designs on spatial prediction and their uncertainties. The background (professional group and frequency of use of statistics) of the information user had no influence in the responses selected for each approach. Of these methods we tested, the offset correlation was most favoured, but had no direct link to decision making and some methods of communication were not well understood (joint probabilities and implicit loss functions). There were consistent responses under the offset correlation–compared to the other methods, and will likely be more useful to information users, with little or no statistical background, who are unable to express their requirements of information quality based on other measures of uncertainty. Although previous work has found that uncertainty of spatial information is best understood when presented in terms of a decision-specific metric, that was not the case here. This shows that more work must be done to develop and elucidate decision specific approaches, perhaps through methods to elicit useful loss functions.

.

## Appendix A: Theory

In Appendix A of this paper, we describe in detail the test approaches.

### A1 Prediction interval

Some unknown quantity at a location (e.g. soil pH or $Se_{grain}$) is characterised by a prediction distribution conditional on the data and statistical model. The kriging prediction is a weighted average of the data

$$\tilde{Z}(\mathbf{x}_0) = \sum_{i=1}^{N} \lambda z(\mathbf{x}_i), \tag{A1}$$

where $z(\mathbf{x}_i)$ is the data and $\lambda$ are the kriging weights (Webster and Oliver, 2007). The kriging variance, $\sigma_{\mathrm{K}}^2$ is defined as:

$$\sigma_{\mathrm{K}}^2 = \mathrm{E}[\{Z(\mathbf{x}_0) - \tilde{Z}(\mathbf{x}_0)\}^2]. \tag{A2}$$

Cross-validation predictions of the statistical model need to be examined by exploratory analysis of the kriging error, $\varepsilon(\mathbf{x}_0)$, defined as $\varepsilon(\mathbf{x}_0) = \{z(\mathbf{x}_0) - \tilde{Z}(\mathbf{x}_o)\}$ to check if the assumption of the normality holds. The kriging predictor is unbiased and the mean of the errors is zero, and their standard deviation is equal to the kriging standard deviation, $\sigma_{\mathrm{K}}$, from kriging. Based on this, a 95% prediction interval can be computed as:

$$\left[\tilde{Z}(\mathbf{x}_0) - 1.96\sigma_{\mathrm{K}}(\mathbf{x}_0), \tilde{Z}(\mathbf{x}_0) + 1.96\sigma_{\mathrm{K}}(\mathbf{x}_0)\right]. \tag{A3}$$

The prediction distribution may also be obtained on a block support–for example if predictions are required at the scale of a farm mean or a mean for an administrative region. The same approach holds to the derivation of a prediction interval.

## A2  Joint probability

We can calculate the joint probability that a location requires an intervention, and that the kriged estimate does not indicate this. If $\mathbf{x}_0$ is the location of interest, $\tilde{Z}(\mathbf{x}_o)$ is the prediction and $z(\mathbf{x}_0)$ the value of the variable at $\mathbf{x}_0$, then $\tilde{Z}(\mathbf{x}_o) - z(\mathbf{x}_0)$ $= \varepsilon(\mathbf{x}_0)$, the error of the kriging predictions. The covariance of $z(\mathbf{x}_0)$ and $\varepsilon(\mathbf{x}_0)$ is:

$$\mathrm{Cov}\left[\mathrm{z}(\mathbf{x}_0), \varepsilon(\mathbf{x}_0)\right] = \mathrm{Var}\left[\mathrm{Z}(\mathbf{x}_0)\right] - \boldsymbol{\lambda}^{\mathrm{T}}\mathbf{c}, \tag{A4}$$

where $\boldsymbol{\lambda}$ denotes the vector of kriging weights for observations used to make the prediction, and $\mathbf{c}$ denotes the vector of covariances between each of these observations and $Z(\mathbf{x}_0)$. We can therefore, specify the joint distribution of $\{z(\mathbf{x}_0), \varepsilon(\mathbf{x}_0)\}$, assuming a normal random variable and prediction errors and conditional on the variance parameters of a geostatistical model. We also specify some $\mathbf{x}_0$ which will give a conservative output–e.g. for a square grid we could specify $\mathbf{x}_0$ at the centre of a grid cell where kriging variance is largest. From this it is possible to compute the joint probability that $\tilde{Z}(\mathbf{x}_0) \geq z_t$ given that $z(\mathbf{x}_0) < z_t$, i.e. the probability, given that an intervention is required at $\mathbf{x}_0$ that, due to error in prediction, the mapped variable does not show this. A detailed description of how the desired conditional probability can be obtained from the joint probabilities is presented in the Supplement.

## A3  Implicit loss function

The loss is a function of the error, if $\tilde{Z}$ is the predicted value and the true value is $z$, then the error is $\mathcal{L}(\tilde{Z} - z)$. If the value of $z$ is equals to 0, then the error is equal to $\tilde{Z}$. The loss function is explained in greater detail by Journel (1984), Goovaerts (1997) and Lark and Knights (2015). Journel (1984) defined a general linear loss function as:

$$\mathcal{L}(\tilde{Z} - z) = \alpha_1|\tilde{Z} - z| \text{ if } \tilde{Z} < z$$
$$\qquad\quad = \alpha_2|\tilde{Z} - z| \text{ if } \tilde{Z} \geq z. \tag{A5}$$

The parameters $\alpha_1$ and $\alpha_2$ have positive real values. The coefficient $\alpha_2$ is the loss per unit error of underestimation and $\alpha_1$ is the loss per unit of error of overestimation. The slopes, $\alpha_1$ and $\alpha_2$ define the asymmetry of the loss function. The loss function can be symmetrical, i.e. penalizing overestimation and underestimation equally; or can be asymmetrical because over-and-underestimation have different consequences. The asymmetry of the loss function is the ratio of the loss per unit value by which a quantity is underestimated to the loss per unit value of an overestimation (Lark and Knights, 2015). The asymmetry,

$a$, is obtained by

$$a = \frac{\alpha_2}{\alpha_1}, \tag{A6}$$

i.e., is independent of the absolute value of $z$. If the loss function depends only on the estimation error, then $z$ can be set to zero, without loss of generality and the expected loss can be computed as a function of the error variance, and so of the sample size (Lark and Knights, 2015). Increasing sample size reduces the minimum expected loss in so far as it reduces the error variance. Therefore, the cost of obtaining $n$ samples can be measured at which the marginal cost of an additional sample point is equal to the reduction in expected loss that single sample achieves (Goovaerts, 1997). However, it maybe difficult to define a loss function prior to making decisions about sampling. The losses may not be easy to quantify, e.g. social costs of failing to intervene, costs of unnecessary interventions, loss of confidence in the decision-making organisation. information users can be helped to reflect on possible loss functions through the implicit loss function. It is a loss function that makes a specified sample size, $n$, a rational choice, given the marginal costs. That is to say, it is the loss function implied by a choice of $\bar{n}$, assuming this is rational. The implicit loss function is conditional on a logistic model, that expresses the marginal costs of the sampling exercise and the conditional distribution of $z$ as a function of effort (Lark and Knights, 2015) and is obtained by finding $\bar{\alpha}_1$ (given asymmetry), such that

$$\breve{\mathcal{L}}(\bar{n}-1|\bar{\alpha}_1,\bar{\alpha}_2,\boldsymbol{\phi}) - \breve{\mathcal{L}}(\bar{n}|\bar{\alpha}_1,\bar{\alpha}_2,\boldsymbol{\phi}) = \mathrm{C}(\bar{n}) - \mathrm{C}(\bar{n}-1), \tag{A7}$$

where $\bar{n}$ is the specified number of samples, $\mathrm{C}(\mathrm{n})$ is the function that returns the cost of $n$ samples and $\boldsymbol{\phi}$ is a vector of variogram parameters, so kriging variance is a contributor. The asymmetry can be set at different values, or inferred from other elicited opinions of the information user group (Lark and Knights, 2015). The expected loss can be minimised at a location given some prediction distribution of $\tilde{Z}$ for the variable of interest by specifying the value of variable corresponding to a given probability ($P_0$), i.e.,

$$\tilde{Z} = F^{-1}(P_0). \tag{A8}$$

Where, $F^{-1}$ denotes the quantile of the prediction distribution for a probability $P_0$ obtained from

$$P_0 = \frac{\alpha_2}{\alpha_1 + \alpha_2}, \tag{A9}$$

(Journel, 1984). Lark and Knights (2015) suggested that a information user group might consider an implicit loss function for different $\bar{n}$ as starting points in the elicitation of a sample size, or compare implicit loss functions for different projects given different partitions of a total budget between them. No attempt has been made to elicit opinions from information users on implicit loss function, so we tried it in this study.

**A4 Offset correlation**

The expected correlation between the kriging predictions, $\tilde{Z}_1(\mathbf{x}_0)$, made from data collected on a square grid, of interval $\zeta$, and predictions, $\tilde{Z}_2(\mathbf{x}_0)$, made from a second grid, a translation of the first grid by $\zeta/2$ in both directions is known as the offset

correlation. The correlation of the two kriging predictions can be computed by:

$$\rho_{\tilde{Z}_1, \tilde{Z}_2} = \frac{\mathbf{C}_{\tilde{Z}_1, \tilde{Z}_2}(\mathbf{x}_0)}{\sqrt{\sigma^2_{K_{\tilde{Z}_1}} \sigma^2_{K_{\tilde{Z}_2}}}}, \tag{A10}$$

where $\mathbf{C}_{\tilde{Z}_1, \tilde{Z}_2}(\mathbf{x}_0)$ is the covariance $\tilde{Z}_1(\mathbf{x}_0)$ and $\tilde{Z}_2(\mathbf{x}_0)$. $\sigma^2_{K_{\tilde{Z}_1}}$ and $\sigma^2_{K_{\tilde{Z}_2}}$ are the kriging variances of the predictions from the first and second grid, respectively.

The offset correlation depends on $\mathbf{x}_0$, and is smallest at the location furthest from points on either grid. This minimum offset correlation is used to evaluate predictions from a grid spacing $\zeta$. As the uncertainty in the map, attributable to sample density, decreases, the offset correlation increases. The denser the grid the more consistent the maps and the offset correlation will be 1 if the maps are identical and 0 if they are entirely unrelated to each other. The offset correlation is bounded on the interval [0,1], and ranges from zero (when the maps produced from the two grids are independent of each other (at a coarse spacing)

and approach 1 as the grid becomes finer and the two maps become increasingly similar. Lark and Lapworth (2013) describes the offset correlation in greater detail.

### Appendix B: Composition of Participants

In this Appendix we present pie charts showing the percentage of participants by (a) location, (b) level of mathematical education, (c) level of use of statistics and (d) professional group

### Appendix C: Contingency tables

In Appendix we described how a contingency table can be partitioned to evaluate whether there are differences in the responses of the participants based on (i) variable used in the test method, (ii) professional group and (iii) by frequency of use of statistics. In Table C1, we illustrate how the contingency table can be partitioned. The table can be partition into components corresponding to pooled table and subtables of the full table.

The full table in Table C1, was partitioned into components corresponding to subtables for soil pH (Subtable 1 in Table C1), and $\mathrm{Se_{grain}}$ concentration (Subtable 2 in Table C1). Then the pooled table completes the partition. The degrees of freedom and deviances for the three table sum to the degrees of freedom and deviance of the full table. Using the contingency table, we could conclude if there are differences in responses for the two variables. The full table in can further be partitioned, in a similar way, by the background of the respondents i.e., professional group and frequency of use of statistics.

The full contingency table for Q1, for offset correlation, is presented as Table C2, in the appendix. The table shows how many individuals selected the given responses for offset correlation. This table is according to variable used (soil pH vs. $\mathrm{Se_{grain}}$), professional group and frequency of use of statistics. Table C3 shows how many individuals selected a given response to Q1, for offset correlation, when columns are pooled within variable used, soil pH or $\mathrm{Se_{grain}}$ concentration. Table C4 shows the pooled counts of the responses for Q1.

**Table C1.** An illustration of how the log-likelihood ratio was used to partition full table into subtables and pooled tables.

**Full table**

Deviance = $L_f$
Response = $O_{i,j}$
degrees of freedom = $DF_F$ = (2−1)×(12−1) = 11

| | Soil pH | | | | | | $Se_{grain}$ concentration | | | | | |
| | Agronomy or soil science (AGS) | | | Public health or nutrition (PHN) | | | Agronomy or soil science (AGS) | | | Public health or nutrition (PHN) | | |
| Response | Occasionally | Regularly | Perpetually | Occasionally | Regularly | Perpetually | Occasionally | Regularly | Perpetually | Occasionally | Regularly | Perpetually |
|---|---|---|---|---|---|---|---|---|---|---|---|---|
| Spacing 1 | $O_{1,1}$ | $O_{1,2}$ | $O_{1,3}$ | $O_{1,4}$ | $O_{1,5}$ | $O_{1,6}$ | $O_{1,7}$ | $O_{1,8}$ | $O_{1,9}$ | $O_{1,10}$ | $O_{1,11}$ | $O_{1,12}$ |
| Spacing 2 | $O_{2,1}$ | $O_{2,2}$ | $O_{2,3}$ | $O_{2,4}$ | $O_{2,5}$ | $O_{2,6}$ | $O_{2,7}$ | $O_{2,8}$ | $O_{2,9}$ | $O_{2,10}$ | $O_{2,11}$ | $O_{2,12}$ |

**Pooled table 1**

Pooled table (Professional groups pooled within variable):
Deviance = $L_{p1}$
degrees of freedom = $DF_{P1}$ = (2−1)×(2−1) = 1

| Response | Soil pH | $Se_{grain}$ concentration |
|---|---|---|
| Spacing 1 | $O_{1,1}+O_{1,2}+O_{1,3}+O_{1,7}+O_{1,8}+O_{1,9}$ | $O_{1,4}+O_{1,5}+O_{1,6}+O_{1,10}+O_{1,11}+O_{1,12}$ |
| Spacing 2 | $O_{2,1}+O_{2,2}+O_{2,3}+O_{2,7}+O_{2,8}+O_{2,9}$ | $O_{2,4}+O_{2,5}+O_{2,6}+O_{2,10}+O_{2,11}+O_{2,12}$ |

**Subtable 1**

Subtable 1 (Soil pH):
Deviance= $L_{s1}$
degrees of freedom = $DF_{S1}$ = (2−1)×(6−1) = 5

| | Soil pH | | | | | |
| | Agronomy or soil science (AGS) | | | Public health or nutrition (PHN) | | |
| Response | Occasionally | Regularly | Perpetually | Occasionally | Regularly | Perpetually |
|---|---|---|---|---|---|---|
| Spacing 1 | $O_{1,1}$ | $O_{1,2}$ | $O_{1,3}$ | $O_{1,4}$ | $O_{1,5}$ | $O_{1,6}$ |
| Spacing 2 | $O_{2,1}$ | $O_{2,2}$ | $O_{2,3}$ | $O_{2,4}$ | $O_{2,5}$ | $O_{2,6}$ |

**Subtable 2**

Subtable 2 ($Se_{grain}$ concentration):
Deviance = $L_{s2}$
degrees of freedom= $DF_{S2}$ = (2−1)×(6−1) = 5

| | $Se_{grain}$ concentration | | | | | |
| | Agronomy or soil science (AGS) | | | Public health or nutrition (PHN) | | |
| Response | Occasionally | Regularly | Perpetually | Occasionally | Regularly | Perpetually |
|---|---|---|---|---|---|---|
| Spacing 1 | $O_{1,7}$ | $O_{1,8}$ | $O_{1,9}$ | $O_{1,10}$ | $O_{1,11}$ | $O_{1,12}$ |
| Spacing 2 | $O_{2,7}$ | $O_{2,8}$ | $O_{2,9}$ | $O_{2,10}$ | $O_{2,11}$ | $O_{2,12}$ |

Deviance partition:
$L_f = L_{p1} + L_{s1} + L_{s2}$

Degrees of freedom partition:
$DF_f = DF_{p1} + DF_{s1} + DF_{s2}$

Table C2. The full contingency table for Q1 for offset correlation, showing how many respondents selected the given responses for offset correlation. The table is according to variable used, professional group and frequency of use of statistics in job role. The figures in parentheses are the expected numbers, $e_{i,j}$ a product of row and column totals dived by the total number of responses. The acronyms represent the professional groups (AGS–agronomist or soil scientist; PHN– public health or nutrition specialists), and frequency of use of statistics in job role (Pep– perpetual use of statistics; Occ–Occasional use of statistics and Reg– regular use of statistics).

| Response | soil pH | | | | | | $Se_{grain}$ | | | | | |
| | AGS | | | PHN | | | AGS | | | PHN | | |
| | Pep | Occ | Reg | Pep | Occ | Reg | Pep | Occ | Reg | Pep | Occ | Reg |
|---|---|---|---|---|---|---|---|---|---|---|---|---|
| Offset=0.4 | 0(0.23) | 0(0.31) | 1(0.85) | 0(0.08) | 0(0.31) | 0(0.23) | 0(0.23) | 2(0.31) | 1(0.31) | 0(0.08) | 0(0.31) | 0(0.23) |
| Offset=0.5 | 0(0.17) | 1(0.23) | 0(0.63) | 0(0.06) | 0(0.23) | 1(0.17) | 0(0.17) | 0(0.23) | 1(0.23) | 0(0.06) | 0(0.23) | 0(0.17) |
| Offset=0.6 | 1(0.40) | 0(0.54) | 1(1.48) | 0(0.13) | 1(0.54) | 0(0.40) | 1(0.40) | 0(0.54) | 0(0.54) | 0(0.13) | 1(0.54) | 2(0.40) |
| Offset=0.7 | 1(0.92) | 2(1.23) | 2(3.38) | 0(0.31) | 3(1.23) | 2(0.92) | 1(0.92) | 1(1.23) | 3(1.23) | 0(0.31) | 1(1.23) | 0(0.92) |
| Offset=0.8 | 0(0.87) | 0(1.15) | 5(3.17) | 1(0.29) | 0(1.15) | 0(0.87) | 0(0.87) | 1(1.15) | 5(1.15) | 1(0.29) | 1(1.15) | 1(0.87) |
| Offset=0.9 | 1(0.40) | 1(0.54) | 2(1.48) | 0(0.13) | 0(0.54) | 0(0.40) | 1(0.40) | 0(0.54) | 1(0.54) | 0(0.13) | 1(0.54) | 0(0.40) |

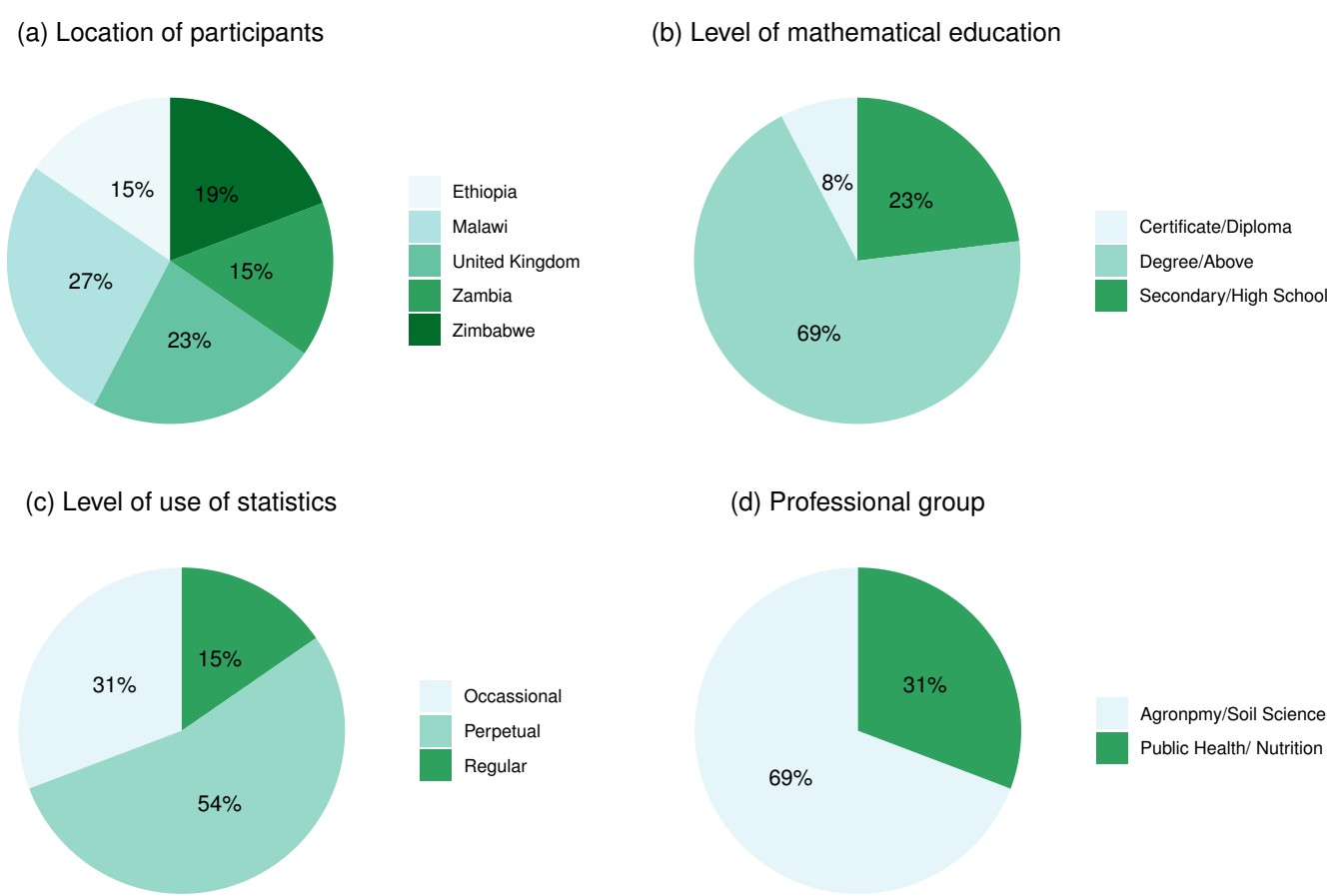

**Figure B1.** Pie charts showing the percentage of participants by (a) location, (b) level of mathematical education, (c) level of use of statistics and (d) professional group.

**Table C3.** A subtable showing how many individuals selected a given response to Q1, for offset correlation, when columns are pooled within variable used (soil pH vs. $Se_{grain}$ concentration).

| Response | soil pH | $Se_{grain}$ |
|---|---|---|
| Offset=0.4 | 1 | 3 |
| Offset=0.5 | 2 | 1 |
| Offset=0.6 | 3 | 4 |
| Offset=0.7 | 10 | 6 |
| Offset=0.8 | 6 | 9 |
| Offset=0.9 | 4 | 3 |

**Table C4.** Pooled responses given to the question on offset correlation.

| Response | Pooled counts |
| --- | --- |
| Offset=0.4 | 4 |
| Offset=0.5 | 3 |
| Offset=0.6 | 7 |
| Offset=0.7 | 16 |
| Offset=0.8 | 15 |
| Offset=0.9 | 7 |

*Author contributions.* The study design was conceived and implemented by CC, RML and AEM. PCN and MRB were responsible for project administration and funding. PCN and JGC supervised the data collection. All authors contributed to the preparation of the article.

*Competing interests.* The authors declare that they have no conflict of interest.

*Disclaimer.* The funders were not involved in the design of this study or the collection, management, analysis and interpretation of the data, the writing of the report or the decision to submit the report for publication.

*Ethics statement.* Ethical approval to conduct this study was granted by the University of Nottingham, School of Biosciences Research Ethics Committees (SBREC202122022FEO) and participants gave informed consent to their participation and subsequent use of their responses.

*Acknowledgements.* This work was funded by the Nottingham-Rothamsted Future Food Beacon Studentships in International Agricultural Development and supported by the Bill & Melinda Gates Foundation [INV-009129]. Under the grant conditions of the Foundation, a Creative
Commons Attribution 4.0 Generic License has already been assigned to the Author Accepted Manuscript version that might arise from this submission.

The authors gratefully acknowledge the contributions made to this research by the participating farmers and field sampling teams from the Department of Agricultural Research Services, and Lilongwe University of Agriculture and Natural Resources.

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
