# Peer review of "Planning a geostatistical survey to map soil and crop properties: communicating uncertainty and its dependence on sampling density to stakeholders"

_Geoscience Communication, 2023_

## Author Comment (AC1)

**Geoscience Communication gc-2023-1 – Reply on RC2**

| Referee Comment | Author Comment |
|---|---|
| The work is nicely thought out and well written, and I think this will be useful to add to the literature about how easy users find it to understand uncertainty when presented in different ways. It also nicely demonstrates that different sampling densities would result if the different methods of communicating the uncertainties would be used. I have only very minor comments to add, plus some typos. | We would like to thank the referees for their thorough review of our manuscript. We wish to revise the manuscript based on their suggestions and comments. We reply to each of the comments below. Our suggested edits in the paper are in blue below, with line numbers indicating where we wish to make the changes. |
| I think the introduction should mention other mapping methods, which can make use of appropriate environmental covariates to model part of the variation, and then justify the focus of the work on kriging and gridded sample designs. If other mapping methods were used, optimal sampling schemes would then not be a grid, could the methods be applied to help in this case? | We have proposed to make changes in the introduction as we were responding to RC1 comments. |
| 14: correlation is bounded on [-1,1]? | Correlations are bounded [-1,1] however, offset correlation ranges from zero (when the maps produced from the two grids are independent of each other (at a coarse spacing) and approach 1 as the grid becomes finer and the two maps become increasingly similar. Therefore, we wish to make the following change in the manuscript from L170 to make it clearer for the reader:

The offset correlation is bounded [0,1], and ranges from zero (when the maps produced from the two grids are independent of each other (at a coarse spacing) and approach 1 as the grid becomes finer and the two maps become increasingly similar. |
| 35: Clarify that different grid spacings is for the data (not grid of predictions). | Suggested edit will be made on the manuscript L35: |

| | Therefore, if we have a reasonable estimate of variance parameters (i.e. variogram for ordinary kriging) we can compute kriging variances for different grid spacings for the data and, in principle, select an acceptable one (McBratney et al., 1981). |
|---|---|
| 42: Should this specifically say "ordinary kriging predictions" (ie not simple/regression/universal kriging) | Suggested edit will be made on the manuscript L35:

The kriging variance, at some location, depends only on the variogram and the spatial distribution of observations for ordinary kriging predictions (Webster and Oliver, 2007; Webster and Lark, 2013). |
| Eq 2: I think z(x0) on right-hand side should be z(xi), also in the text below the equation. | Suggested edit will be made on the manuscript L100 on Equation 2:

$$\tilde{Z}(\mathrm{x_0}) = \sum_{I=1}^{N} \lambda z(\mathrm{x_i}),$$

where $z(\mathrm{x_i})$ is the data and $\lambda$ are the kriging weights (Webster and Oliver, 2007). |
| Eq 3: This is a repeat of Eq 1. Probably better here (ie after presenting formula for prediction), so suggest deleting Eq 1. | We propose to delete the following sentence on L97 and equation 1, as suggested by the referee:

 |
| 103: definition of epsilon here should align with what is given on line 112 (probably line 112 version is better). | Suggested edit will be made on L103:

Cross-validation predictions of the statistical model need to be examined by exploratory analysis of the error of the kriging prediction, $\varepsilon(x_0)$, defined as $\{\tilde{Z}(x_0) - z(x_0)\}$, to check the if the assumption of normality holds. |
| 134-136: I'm confused by this sentence. If z is 0 in Eq 6, then the true data does not appear in the equation, and the error doesn't seem to matter, only the value of the prediction? | In the equation the loss is a function of the error, if Z* is the predicted value and the true value is z, then the error is (Z* - z).  If the true value is 0 then the error is equal to Z*, that is not the same as saying that the data value has vanished from the error. |

| | |
|---|---|
| 162: I think it would be clearer to put "made from data collected on a square grid" | Suggested edit will be made on L162: |
| | The expected correlation between the kriging predictions. $\tilde{Z}_1(x_0)$, made from data collected on a square grid, of interval $\zeta$, and predictions, $\tilde{Z}_2(x_0)$, made from a second grid, a translation of the first grid by $\zeta/2$ in both directions is known as the offset correlation. |
| 230: Was the size of blocks the same for all grids, or was it set to be the same size as the grid? The second option (block size = grid size) doesn't really make sense to me in this context, as the meaning of the predictions (and their variances) would be different for the different sample designs. | We used the same size of blocks for all the grids, and it was 0.01 km² square block, and we wish to make the following change in the manuscript on L230. |
| | We then computed the cell-centred block kriging variance the spacings we were considering by block kriging (Webster and Oliver, 2007). For all the grid spacings, we computed cell-centred block kriging variance on 0.01 km² square blocks. |
| Fig 4: Can brief details of how these maps be added? Were data for a pair of offsetted sampling grids jointly simulated, then those simulated data (for each of the two offsetted grids in turn) used to predict on the fine-scale mapping grid (as shown in the figure)? | The maps were generated to allow the participant to visualize what two spatial variables correlated by some specified amount would look like. They were generated as realizations of two coregionalized variables with the same mean and variance and the target correlation specified. |
| 450: Needs a sentence added here to summarise what you did (before talking about responses in the next sentence). | We have added the following sentence from L450 as suggested by the referee: |
| | In this study we evaluated four methods of communicating uncertainty associated with kriging predictions made from data from a geostatistical survey, to determine an appropriate sampling density to meet stakeholders expectations. |
| Tables A2 and A3: these could easily be combined into one table | Tables A2 shows the subtable for responses when pooled within the variable used (soil pH and Se$_{grain}$) when partitioning the full table as illustrated on Table 2. Then table A3 shows the pooled counts of response for offset correlation after all the partitioning for Q1, and this was used to examine if the responses were uniformly distributed. |

| Line number: suggested new text: | Suggested edit will be made on the manuscript L7: |
|---|---|
| 7: "from data on sample grids…" | Offset correlation is a measure of the consistency of kriging predictions made from data on sample grids with the same spacing but different origins |
| Line number: suggested new text:

20: "concentrations" | Suggested edit will be made on the manuscript L20:

In the GeoNutrition project, it has been shown that concentrations of micronutrients in staple crops and in soils vary spatially and so interventions to address the deficiencies should be based on spatial information (Gashu et al., 2021; Botoman et al., 2022). |
| Line number: suggested new text:

26: "survey efforts" | Suggested edit will be made on the manuscript L26:

Often survey efforts are constrained by budgets and we need a trade-off between sample effort and reducing uncertainty. |
| Line number: suggested new text:

33: "is quantified" | Suggested edit will be made on the manuscript L33:

In geostatistical prediction, the variogram function models the spatial dependence of the variable of interest, and the uncertainty in the predicted values is quantified by the kriging variance (i.e., the mean squared error of the prediction). |
| Line number: suggested new text:

53: "tied to particular decisions" | However, we know that prediction intervals are not preferred by end-users as a method of communicating uncertainty when making decisions, they find it easier to interpret measures of uncertainty tied to particular decisions (Chagumaira et al., 2021). |
| Line number: suggested new text:

61: delete comma after x0 | Suggested edit will be made on the manuscript L61:

For a conservative measure of uncertainty, $x_0$ may be at a general location where uncertainty is largest e.g., at the centre of a square grid cell. |
| Line number: suggested new text: | Suggested edit will be made on the manuscript L67: |

| | |
|---|---|
| 67 "to an acceptable" | The conditional probabilities can then be used to make a decision about soil sampling, by selecting an appropriate grid spacing which limits the risk to an acceptable level. |
| Line number: suggested new text:

79: "or from a comparable region" | Suggested edit will be made on the manuscript L79:

The logistical model can be obtained from data from a previous survey or from a comparable region. |
| Line number: suggested new text:

138: "reduces the error" | Suggested edit will be made on the manuscript L138:

Increasing sample size reduces the minimum expected loss in so far as it reduces the error variance. |
| Line number: suggested new text:

139: "an additional sample point" | Suggested edit will be made on the manuscript L139:

Therefore, the cost of obtaining $n$ samples can be measured at which the marginal cost of an additional sample point. |
| Line number: suggested new text:

145: "the sampling exercise" | Suggested edit will be made on the manuscript L145:

The implicit loss function is conditional on a logistic model, that expresses the marginal costs of the sampling exercise |
| Line number: suggested new text:

149: "number of samples" | Suggested edit will be made on the manuscript L149:

where $\bar{n}$ is the specified number of samples, C(n) is the function that returns the cost of n samples and $\phi$ is a vector of variogram |
| Line number: suggested new text:

155: "denotes" | Suggested edit will be made on the manuscript L155:

Where, $F^{-1}$ denotes the quantile of the prediction distribution for a probability $P_0$ obtained from |
| Line number: suggested new text:

158: "loss functions" | Suggested edit will be made on the manuscript L158:

Lark and Knights (2015) suggested that a stakeholder group might consider an implicit loss function for different $\bar{n}$ starting points in the |

| | elicitation of a sample size or compare implicit loss functions for different projects given different partitions of a total budget between them. |
|---|---|
| Line number: suggested new text:

231: "three different predictions" | Suggested edit will be made on the manuscript L231:

We considered three different predictions for each variable, but the prediction interval was fixed, depending only on grid spacing. |
| Line number: suggested new text:

244: "asked the participant what grid" | Suggested edit will be made on the manuscript L244:

We then asked the participant what grid spacing they thought corresponded to the largest acceptable value of this probability. |
| Line number: suggested new text:

267: "pair of maps" | Suggested edit will be made on the manuscript L267:

Figure 4, shows an example of pair of maps of Se$_{grain}$ concentration and the corresponding scatterplot (see Figure S5 and S6). |
| Line number: suggested new text:

289: "partitioned into components corresponding to a pooled table".
291: I think "Figure 3" should be "Table 2" | Suggested edit will be made on the manuscript L289 and L291:

The full table in Table 2, was partitioned into components corresponding to subtables for soil pH (subtable 1 in Table 2), and Segrain concentration (Subtable
2 in Table 2). |
| Line number: suggested new text:

302: "differences in responses" | Suggested edit will be made on the manuscript L302:

We first tested for differences in responses recorded for each test method, by the variable used (soil pH or Se$_{grain}$ concentration) using contingency tables. |
| Line number: suggested new text:

311: "no difference" | Suggested edit will be made on the manuscript L311:

For some questions, we noted differences in the responses when pooled within variable used (soil pH or Segrain concentration) and there was no difference in responses in professional groups and frequency of use of statistics for all |
| Line number: suggested new text: | Suggested edit will be made on the manuscript L312: |

| | |
|---|---|
| 312: "were uniformly" | We further analysed the pooled tables or separate subtables to examine if the responses were uniformly distributed and the null hypothesis is a random distribution. |
| 329: "responses of the?" Missing something here. | We have made the following edit on L329:

There were no differences in the responses when the columns were pooled by the variable used, soil pH vs. Segrain concentration, p = 0.656 (Table 3). |
| Line number: suggested new text:

372: I don't think this should be "by all respondents" (ie not every single respondent ranked it first?), maybe should be "Amongst all respondents, the offset…most effective" | Suggested edit will be made on the manuscript L372:

Amongst all the respondents, the offset correlation was ranked as the most effective (Figure 9a) and implicit loss function as the least effective. |
| Line number: suggested new text:

380: "statistics" | Suggested edit will be made on the manuscript L380:

Those who always use statistics, ranked conditional probabilities second. |
| Line number: suggested new text:

388: "explains why there were" | Suggested edit will be made on the manuscript L388:

This explains why there were more consistent responses under this method |
| Line number: suggested new text:

404: "This suggests" | Suggested edit will be made on the manuscript L404:

This suggests that the stakeholders may not have fully understood the method. |
| Line number: suggested new text:

417: "by the respondents" and "would be of greatest value" | Suggested edit will be made on the manuscript L417:

Similar reasons were given by the respondents. We expected that prediction intervals would be of greatest value for specific interpretation of particular sites but would be of limited value for survey planning. |

| Line number: suggested new text: | Suggested edit will be made on the manuscript L432: |
|---|---|
| 432: "beginning" and "explanation of" | At the beginning of the online workshop, we explained each method with the aid of illustrations. After an explanation of each method, there was a feedback session to allow the participants opportunities to seek clarity on ambiguous and unfamiliar concepts from the presenters. |
| Line number: suggested new text:

441: "different variables" | Suggested edit will be made on the manuscript L441:

All the methods may give different results for different variables, because they depend on the variogram of the variable in question. |

---

## Author Comment (AC2)

**Geoscience Communication gc-2023-1: Reply on RC1**

| Referee Comment | Author Comment |
|---|---|
| | We would like to thank the referee for their thorough review of our manuscript. We wish to revise the manuscript based on their suggestions and comments. We reply to each of the comments below. Our suggested edits in the paper are in blue below, with line numbers indicating where we wish to make the changes. |
| (i) The shift from the goal of science communication towards complex mathematical modelling leaves me perplexed regarding several key aspects. | (i) It is precisely this shift between complex model output and communication which is the concern of this paper. Other papers we have published (Chagumaira et al., 2021, 2022) have addressed the question of how the output of such models, which quantifies the uncertainty of spatial predictions, can be communicated to users of the information. In this paper we recognize that one of the strengths of the geostatistical modelling process which statisticians have exploited is that the uncertainty of predictions can be proposed a priori for different sampling intensities, which can help when deciding how much effort to put into field work. However, this only works if the criteria for information quality can be communicated effectively to all stakeholders as the decision on survey effort is one which must be made collaboratively, the final decision resting with the survey sponsor or science lead who typically might not have statistical expertise. That is what we address in this paper. When translated effectively, mathematical models are powerful tools for engaging diverse audiences, and explore different scenarios and understand the cause-and-effect relationships within geosystems. |

To clarify this for the reader we propose the following edit of the introduction from line 24 onward. This contains additional material included to address the reviewer's point about technical detail in the paper.

[revised manuscript text omitted]

Apart from the presentation style to the participants, it's crucial to recognize that the map quality presented to stakeholders is influenced by multiple variables, such as the initial sampling density, data quality, sampling design, and robustness of the geostatistical models themselves. The paper's direction, asking stakeholders to rank methods based on their effectiveness, seems ill-defined and lacking theoretical grounding. Furthermore, the framing of the quality of input data solely as a function of sampling density is simplistic and ignores other essential qualitative considerations, such as the type of data collected. While the decision on sampling density is undoubtedly vital, the assumption that it should be taken a priori disregards other vital factors like the scale, sampling strategy (design), and the level of detail of the phenomenon under study. These become paramount when assessing the overall quality of the required output. | (i) The reviewer refers to hypothetical map pairs designed to allow the respondent to visualize what two spatial variables with a certain correlation might look like. This is in the specific context of the offset correlation measure. We propose to edit the text at L265 to read "

We presented the participants with correlated pairs of hypothetical maps, with differing correlations, so that the extent to which maps might differ as a result of the grid offset could be visualized. |
| A particularly concerning aspect of this paper is the scant information provided about the stakeholders' selection, background, and representation. Grouping of soil scientists and agronomists together, for instance, and juxtaposing them with public health experts and nutritionists, lacks clear justification. | Stakeholders with different disciplines need to work together on complex problems such as MND, and it makes sense to involve them all in the process. As the elicitation obtained individual responses, the background of each was recorded and accounted for in the analysis (see, for example, Table 2) we were able to assess any differences in understanding associated with educational background, training, and experience in different disciplines. This is an essential element of understanding for our work. We wish to make the following change L190, to clarify this.

This study was conducted with information-users who have been involved with the GeoNutrition project (http://www.geonutrition.com/), which examined strategies to alleviate micronutrient deficiencies (MNDs) in |

| | |
|---|---|
| | Ethiopia and Malawi and included surveys to provide baseline information on MN concentrations in staple crops and soils, and soil properties (such as pH) which influence soil to plant transfers of MN. The GeoNutrition project had teams from multiple disciplines (agriculture, soil science, human nutrition, and public health). It has been shown that concentration of micronutrients in staple crops and in soils vary spatially, as do biomarkers for MN status and so interventions to address the deficiencies should be based on spatial information on all these variables (Gashu et al., 2021; Botoman et al., 2022). The spatial information therefore has to be interpreted by information users from this broad set of disciplines, and all of them might also contribute to decisions on the amount of effort to be expended on field survey.  It is plausible that experts with training in different disciplines might find different quantitative methods to express uncertainty in information useful for decision-making, and so we recruited a panel for elicitation which spanned these disciplines. We recruited the panel from institutions which were partners of the GeoNutrition project research team and the allied Translating GeoNutrition project in Zimbabwe (ZimGRTA) and the University of Zambia. These included agricultural research and extension services, public health bodies and nutritional research institutions.  Soil scientists from the UK were also included. Panel members were invited by email from the local GeoNutrition/ZimGRTA lead. |
| The omission of machine learning and AI algorithms, especially in an era defined by big data, further complicates the study's scenario of understanding uncertainties. | Machine learning and AI are important topics in digital soil mapping. However, because they do not entail a statistical model, they provide no basis for rational decisions on sampling intensity. Some work has been done on sample design for ML-based mapping, but these are purely heuristic methods which do not allow sample density to be linked to the quality of the resulting predictions. As we note at L36 (new edited introduction on comments above) our approach is entirely compatible with statistical methods for spatial prediction which use any of the "big data" sources deployed in digital soil mapping |

**Abstract:**

The abstract begins with an unconventional approach, dedicating over five lines to general statements (L1-5). This choice leads to a lack of specificity in addressing the real problem of communicating uncertainty during the planning stage of a geostatistical survey. While there is no scientific dispute that sampling density correlates with prediction uncertainty, this paper fails to elucidate what sets it apart from existing knowledge. There's an opportunity to articulate unique perspectives or new insights on uncertainty, but the paper does not seize it. The introduction of four different ways in which "the relationship between sample density and the uncertainty of predictions can be related" falls short of justifying this research, as no novel insights or values are identified. The abstract would benefit from a more concise focus on the specific problem at hand and a clear rationale for why the chosen methodology is innovative or necessary. Without these clarifications, the abstract's approach feels redundant and fails to engage the reader in a meaningful way.

L8-9: "All four of these methods were investigated using information on soil pH and Se concentration in grain in Malawi" à Investigated in what sense? Please be specific. Additionally, the term "stakeholders" is used in an overly generic way, particularly in the abstract. This lack of specificity leaves the reader wondering who exactly these stakeholders are. Without understanding their roles, experiences, backgrounds, and locations, it's challenging to gauge the relevance and applicability of their opinions and decisions in the context of the research. The paper would benefit greatly from identifying these stakeholders more precisely. Are they soil scientists, agronomists, public health experts, or nutritionists? What qualifies them to contribute to this particular study? The clarity on these questions would not only strengthen the abstract but also establish a solid foundation for the rest of the paper. My concerns regarding the selection, experience, and location of these stakeholders will be elaborated further in the subsequent sections of this review.

We thank the referee for their comments and suggestions. We have restructured the Abstract in the following way:

Much research has examined communication about uncertainty in spatial information to users of that information, but an equally challenging task is enabling those users to understand measures of uncertainty for surveys of different intensity (and so cost) at the planning stage. While statisticians can relate sampling density to measures of uncertainty such as prediction error variance, these do not necessarily help stakeholders (e.g., agronomists, soil scientists, policy makers and health experts) to make rational decisions on how much budget should be assigned to field sampling to produce information of adequate quality. In this study, we considered four ways to communicate uncertainty associated with predictions made based on data from a geostatistical survey, to determine an appropriate sampling density to meet stakeholders expectations. These include two methods based on the conditional prediction distribution: the width of prediction intervals, and the joint probability that a particular intervention is required at a random location, but the spatial information indicates the contrary. A third method, the offset correlation is a measure of the consistency of kriging predictions made from data on sample grids with the same spacing but different origins. The implicit loss function is a method which allows the user to reflect on the valuation of losses from decisions based on uncertain information implicit in selecting some arbitrary sampling density. Evaluation of the four communication methods was done through a questionnaire by eliciting opinions of participants with experience in planning surveys, about the method's comprehensibility and effectiveness and the sampling density that they would select based on that method. Our results show significant differences in how the participants responded to the methods, with the joint probability and implicit loss function approaches being not well understood, and offset correlation was the most understood. During feedback sessions, the stakeholders highlighted that they were more familiar with the concept of

correlation, with a closed interval of [0,1] and this explains the more consistent responses under this method. The offset correlation will likely be more useful to stakeholders, with little or no statistical background, who are unable to express their requirements of information quality based on other measures of uncertainty.

1 Introduction:

The introduction of the paper provides a broad overview of the study's themes, but it appears superficial and lacks a specific focus on the paper's actual subject. Instead of honing in on the unique issue this research aims to address - namely, how stakeholders deal with uncertainty in planning mapping surveys - the section tends to wander through various unrelated topics. For instance, the introduction's first paragraph begins with a discussion of MND in sub-Saharan Africa, a detail that seems incongruent with the non-location-specific context of the research. A considerable portion of the content here is overly generic and fails to pinpoint the problem the study seeks to explore. Furthermore, the paper makes a convoluted attempt to rationalize the methods (such as offset correlation, implicit loss function, kriging variance, conditional probability) presented to the stakeholders. These methods are described in laborious detail, yet the rationale for their comparison remains unclear. The text also fails to address whether there is scientific consensus on the superiority of any one method. Much of this section is bogged down with technical details that may be unengaging for a wider audience. Simplifying some of these concepts would make them more accessible and align better with the journal's target readership. For example, the sentence '… an implicit loss function, conditional on a logistical model (i.e., a function of sampling effort and statistical information about the estimates of the cost of errors) can be modelled as the loss function that makes a particular decision on sampling effort rational (Lark and Knights, 2015)' is impossible to parse.

We thank the referee for the opinion about the structure of the introduction. In response to the referee's comment on the first paragraph, we wanted to give context of how geostatistical mapping has been used as a tool to provide information about soil and crop micronutrient properties in relation to mineral micronutrient deficiencies in sub-Saharan Africa. Studies conducted in this region provided evidence that concentration of micronutrients vary spatial and spatial information is important to design efficient interventions. Probably this background information should have been provided in the methods section, and we wish to move this text to Method section 3, and expand on it, to give this concise background and the rationale why we grouped soil scientists, agronomists, and public health experts.

We will revise this section to reflect the views of the referee, and we already have presented the proposed revisions for the introductions (from L24) in the first response to this referee comments (RC1).

| | |
|---|---|
| The paper's emphasis on the intuitiveness and simplicity of the offset correlation method is presented as an advantage. However, this approach seems to oversimplify the complexities involved and may even have biased the stakeholders towards this method. The way the study was constructed raises concerns that the stakeholders might have been subtly steered towards favouring the offset correlation method. As already indicated, I suggest that a more engaging introduction is written underlining the theoretical foundations of the study and providing compelling justification for study's approach. | |
| 2 Theory:

This section, as it currently stands, does not appear to add significant value to the overall paper. Although some readers might find it informative, its current content might be better suited for an appendix. I recommend relocating this material and replacing it with a comprehensive literature review that outlines the current state of knowledge regarding decision-making in soil and plant surveys. This could include case study examples highlighting the tangible costs incurred by stakeholders who failed to adequately plan and make informed decisions prior to their survey efforts. Further, it would be enlightening to specify the types of stakeholders you have in mind for this research. Detailing their background and roles will help readers better understand how their specific attributes might influence their decision-making process. By making these adjustments, you can create a section that not only maintains the reader's interest but also lays a more robust foundation for the arguments and findings presented later in the paper. | We acknowledge the referee's comment, and we will move this text to the Appendix for the benefit of readers for whom the mathematical content is of limited interest. |
| **3 Materials and methods**

**3.1 Basic approach** | |

| (i) | L176: "We used the four methods, described above, to assess uncertainty in relation to sampling density, considering the problem of measuring a soil property relevant to crop management: soil pH, and a property of the crop: Se$_{grain}$ concentration." àI am not sure what is meant here with assessing uncertainty in relation to sampling density. Also, what "problem" is there when measuring a soil property relevant to crop management? And why specifically soil pH? And Se? Providing this context will not only enhance the reader's understanding but also reinforce the motivation behind the study, making it easier to follow the progression of the research and its significance within the broader scientific landscape. L177-180: "We used variograms from a national survey in Malawi for each variable (Gashu et al., 2021) to obtain sampling densities for further notional sampling for an administrative district in Malawi, Rumphi District, with an area of 4769 km2. The outputs were presented to participants". à While the paper draws on the dataset from Gashu et al. (2021), further details on how this dataset was collected, along with the rationale behind its selection, would strengthen the connection between the data and the study's objectives. Specifically, it is essential to explain the methodology used in collecting the dataset, including how the parameters of the variograms were selected to derive the sampling densities. This information will provide readers with a clear understanding of the data's reliability and relevance to the study. The paper should address potential biases that could arise from using variograms of national level data to derive regional sampling densities, especially considering the comparison of four different methods. Are there similar machine learning approaches? This section must articulate the steps taken to minimize biases, ensuring that all four methods | (i) | We thank the review for these comments to improve our paper and we wish to edit our manuscript to reflects the points raised by the review. We wish to revise the section on the Materials and Method in the text below. We will add information about why we used the data from the GeoNutrition project and how it was collected and whether this was adequate to support predictions at regional level (see proposed changes below). We described in detail how the variograms were modelled in Section 3.2.1 – where we described statistical modelling and spatial predictions of grain Se and pH. Please see proposed revision for the methods section below. |
|---|---|---|---|

were optimal for the input dataset. Providing a context for how the study's scenario would apply to stakeholders needing to understand uncertainty without national-level data will help readers gauge the broader applicability of the findings. Further, clarity on how the output was presented to the participants, whether through PowerPoint, poster format, or other means, and the order of presentation is crucial. These factors could significantly influence participants' understanding and choices and acknowledging them in the paper will enhance the transparency of the process. By addressing these points, the paper can offer a more comprehensive and clear understanding of the data, methods, and process, enhancing both its scientific rigor and accessibility to a broader audience.

| | | | |
|---|---|---|---|
| (ii) | L180: "The participants considered each method in turn and were asked to select a sampling grid density based on the method. After doing this they were asked, for each method: Has the method helped you assess the implication of uncertainty in spatial prediction in as far as it is controlled by sampling? They were then asked: Which of these methods was easiest to interpret? Finally, the participants were asked to rank the method in terms of ease of use. Evaluation of the test methods were done using an online questionnaire on Microsoft Forms" à How! Which aspect of the methods were considered? Was the quality of the method with regards to the output or which specific aspect? On the question of "easier to interpret", how do authors define "easier"? This question is loaded with so much subjectivity that without a clear unbiased scale of what "easy" means, it is impossible to derive any meaning from their answers. | (ii) | The list of specific questions used to elicit stakeholder are listed in Table 1.These questions were sufficient for the participants to understand for example: "We show you here some pairs of examples map of soil pH/Se grain, each pair being based on a different grid spacing, and so, with a different offset correlation. We also show scatter plots which illustrate the strength of the correlation. What do you think is the smallest correlation that would be acceptable if one of the maps were to be used to make decisions?" We think this is clear enough question which prompts critical thinking about the smallest correlation they deemed acceptable to decide. However, we have edited the text so that it becomes clearer to the reader (see proposed changes below). |

| | | | |
|---|---|---|---|
| (iii) | L187-195: "The invited participants self-identified as (i) agronomist or soil scientist or (ii) public health or nutrition specialists. The participants also self-assessed their statistical/mathematical background and their frequency of use of statistics in their job role (perpetual, regular, occasional use)". à Given that this information is one of the pillars of your findings in this work, I wonder why there wasn't the attempt to standardize the backgrounds of the participants. For instance, what qualifies one as any of the professions (agronomist, soil scientist, nutritionist, and public health specialist). Is it based on education level, years of practice, specific training, or other criteria? Was there a reason why such experts were chosen? Do these experts typically have training in interpreting uncertainty in maps? Elaborate on why the distinctions among these professionals were used as the basis for the response. Address whether the 26 participants were intended to represent a broader population or if they were selected for specific reasons. Justify the choice of only 26 participants for this study. Explain why this number was deemed sufficient, considering the scope and objectives of the research. If the sample size is indeed small, acknowledging its limitation and potential biases will improve the rigor of the study. | (iii) | We will add the following text from L477 to address the issue raised by the referee:

All the stakeholders recruited in this study were employed in public sector institutes in roles (e.g., universities, civil organisations, research, and extension) and had experience in their respective fields in an SSA setting. In terms of sample size, we have no prior basis to select a sample size because, as this was the first study of this topic, it was not clear how to select an appropriate effect size. As a result, our major consideration was recruiting individuals willing to participate and with experience in their respective institutions. We therefore attempted to recruit the entire set of suitable respondents in each country. In future work initial power analysis might be considered. |
| (iv) | L195-200: "In the exercise, an introductory talk was given to explain the study's objectives. During the talk, we explained the four test methods (offset correlation, prediction intervals, conditional probabilities and implicit loss function) and how they can be used to assess the implications of uncertainty in spatial predictions to determine appropriate sampling grid space for a geostatistical survey. We explained the structure of the questionnaire to the participants. We emphasized to the participants that we were not testing their | (iv) | Our participants were stakeholders in that they had an interest in being better able to contribute to the planning, execution, and interpretation of surveys to address MND. They were volunteers, recruited from national-level institutions with responsibility for interventions and policy, they were familiar with the GeoNutrition project and so were aware of the importance of being able to engage with the process. They gave informed consent to participate in the elicitation. No remuneration was offered, but all |

mathematical/statistical skills and understanding but rather were testing the accessibility of the methods using their responses"à What drove the participants to engage in the exercise? Understanding their motivations can shed light on the relevance of their input and the validity of their responses. Were they incentivized in any way? Did they have personal or professional interests in the outcome? The term "stakeholder" typically implies an individual or group with a vested interest in the outcome of a particular process or decision. In this context, it remains unclear if the participants indeed stood to gain or lose anything from the exercise. If they did not have a direct stake in the findings or implications of the research, using the term "stakeholder" might be misleading. An explanation or justification for this terminology would enhance the clarity and precision of the paper.

participants in African countries who were not able to participate from institutional offices were provided with a one-day data bundle to allow them to join online.

(v)     L205-210: "The offset correlation was the first method presented to the participants. This was followed by prediction intervals and conditional probabilities. The implicit loss function was the final method presented to the participants. We started with a measure we thought all our stakeholders would most easily understand and then moved on to the more complex methods." à The presentation of the offset correlation method within the research design appears to have been conducted in a manner that may have inadvertently favored this approach. Was there any randomization in how the different methods were presented to the participants? If the offset correlation was consistently presented first, or in a way that highlighted it more prominently, this could influence participants' perceptions and evaluations. Were all the methods described with equal clarity and neutrality? Any differences in language, emphasis, or complexity might have created an uneven playing field,

(v)     It is easy to "blind" in an experiment when you are giving a subject one of two indistinguishable pills, but hardly relevant to this case.

The proposed edits to address the comments raised by the referee are below:

**3. Materials and Methods**

This study was conducted with information-users who have been involved with the GeoNutrition project (http://www.geonutrition.com/), which examined strategies to alleviate micronutrient deficiencies (MNDs) in Ethiopia and Malawi and included surveys to provide baseline information on MN concentrations in staple crops and soils, and soil properties (such

leading participants to gravitate toward the offset correlation method. Was there any attempt to control or assess the potential for bias in how the methods were presented and evaluated? Implementing and reporting on measures such as blinding or counterbalancing could strengthen the credibility of the results. Were participants' preconceived notions or preferences regarding these methods assessed or controlled for? Their prior knowledge or beliefs could also contribute to a bias in their evaluations. Addressing these questions would help to ascertain whether the apparent favoring of the offset correlation method is a genuine reflection of its merits or a product of the research design. A robust examination of these concerns would enhance the rigor and validity of the findings, ensuring that the conclusions drawn are founded on an unbiased assessment of the methods in question.

as pH) which influence soil to plant transfers of MN. The GeoNutrition project had teams from multiple disciplines (agriculture, soil science, human nutrition, and public health). It has been shown that concentration of micronutrients in staple crops and in soils vary spatially, as do biomarkers for MN status and so interventions to address the deficiencies should be based on spatial information on all these variables (Gashu et al., 2021; Botoman et al., 2022). The spatial information therefore must be interpreted by information users from this broad set of disciplines, and all of them might also contribute to decisions on the amount of effort to be expended on field survey. It is plausible that experts with training in different disciplines might find different quantitative methods to express uncertainty in information useful for decision-making, and so we recruited a panel for elicitation which spanned these disciplines. We recruited the panel from institutions which were partners of the GeoNutrition project research team and the allied Translating GeoNutrition project in Zimbabwe (ZimGRTA) and the University of Zambia. These included agricultural research and extension services, public health bodies and nutritional research institutions. Soil scientists from the UK were also included. Panel members were invited by email from the local GeoNutrition/ZimGRTA lead.

Due to the importance of spatial information, we sought to explore future scenarios whereby other countries in sub-Saharan Africa would like to do a similar project would undergo sampling considering lessons the GeoNutrition project. We wanted to determine how best to help end-users (such as those identified in the GeoNutrition project) can best helped to make decisions on crop and soil sampling using data from a prior survey. We therefore used data from the GeoNutrition project, crop and soil properties were measured at national scale in Malawi.

In this survey, field sampling was undertaken to support the spatial prediction of micronutrient concentration in crops and soil across Malawi.

The sampling design was selected to achieve spatial coverage and used 'main-site' and 'close-pair' sampling to support the estimation of variance parameters of the linear mixed model (Lark and Marchant, 2018). The location of sample points were the centroids of the Delauny polygons, resulting from the stratification function in the spcosa library for the R platform (Walvoort et al. 2010). The sample support (0.1 ha circular plot) for the data consisted of bulk soil and grain samples from aliquots within a single field (Gashu et al., 2020). Therefore, the uncertainty quantification of the predictions relates to the mean values of the target variable across such as support within a field at a specified location, and this is appropriate for deriving regional sampling densities. Details about field data collection in Malawi are presented by Gashu et al. (2021), Botoman et al. (2022) and Kumssa et al. (2022). Grain and soil samples were prepared and analysed using methods described in Gashu et al., 2021.

We used variograms for soil pH and $Se_{grain}$ to obtain sampling densities for further notional sampling for an administrative district in Malawi, Rumphi District, with an area of 4769 $km^2$. The outputs were presented to participants in poster format through PowerPoint, and examples of the posters are shown in Figs. S5 – S10 in the Supplement. Ethical approval to conduct this study was granted by the University of Nottingham, School of Biosciences Research Ethics Committees (SBREC202122022FEO) and participants gave informed consent to their participation and subsequent use of their responses.

**3.1 Format of the exercise**

We wanted to elicit from stakeholder the usefulness of proposed methods (offset correlation, prediction intervals, conditional probabilities) in helping them assess the implications of uncertainty in spatial prediction in as far as this is controlled by sampling, considering the problem of measuring a soil property and a micronutrient from a crop. Soil pH and concentration of Se

in grain were used as examples for this case study. We invited professionals working in agriculture, nutrition and health at civic organisations, universities, government departments from Ethiopia, Malawi and wider GeoNutrition sites (United Kingdom, Zambia, and Zimbabwe). In total we had 26 participants (18 were agronomists or soil scientists and 8 public health or nutrition specialists).

The elicitation was conducted online using \cite*{Zoom} in two sessions, $26^{\rm{th}}$ and $28^{\rm{th}}$ April 2022. There were two sessions to accommodate participants from different time zones, and to manage the participants in smaller groups to allow for questions and feedback. The invited participants self-identified as (i) agronomist or soil scientist or (ii) public health or nutrition specialists. The participants also self-assessed their statistical/mathematical background and their frequency of use of statistics in their job role (perpetual, regular, occasional use).

In the exercise, an introductory talk was given to explain the study's objectives. During the talk, we explained the four test methods (offset correlation, prediction intervals, conditional probabilities, and implicit loss function) and how they can be used to assess the implications of uncertainty in spatial predictions to determine appropriate sampling grid space for a geostatistical survey. We explained the structure of the questionnaire to the participants. We emphasized to the participants that we were not testing their mathematical/statistical skills and understanding but rather were testing the accessibility of the methods using their response.

The participants considered each method in turn and were asked to select a sampling grid density based on the method. Evaluation of the test methods was done through a questionnaire, as shown on Table 1. Using the first four questions, Q1 to Q4, we wanted to find out if the method

helped to identify a sampling grid spacing. On Q5, we wanted the participants to
assess the test methods in terms of their effectiveness in finding an appropriate grid spacing. We asked the participants to rank these methods in an order of their effectiveness, in their experience, and in terms of finding a level of uncertainty that they were able to tolerate when deciding about a sampling grid spacing. We asked them to put rank 1 as the most effective method and rank 4 the least. The participants recorded their responses using and online questionnaire on Microsoft Forms.

The offset correlation was the first method presented to the participants. This was followed by prediction intervals and conditional probabilities. The implicit loss function was the final method presented to the participants. We started with a measure we thought all our stakeholders would most easily understand and then moved on to the more complex methods.

**3.2 Test methods**

(i)     Most of the information in 3.2.1 largely repeats the information in 3.1. Thus, I suggest to fuse the information here with that of the section 3.1. I think some of the questions I raised in 3.1 is answered here so I suppose it makes it easy to fuse them. While it is common to cite previous studies for established methods or data, in this case, where the dataset is central to the analysis, it may be beneficial to provide specific details rather than merely referring to other works. For instance, it is unexplained how soil pH and Segrain is measured. This will give readers a more comprehensive understanding of the methods and rationale behind the chosen measurements.

(i)     Section 3.1 is an overview of the key experimental work, the engagement with the participants. Section 3.2 describes our analysis of the data sets and production of the outputs for the participants to use. We think it important to keep these quite distinct, and do not think that there is more than a superficial overlap. Also, we do not believe that the analytical methods are of special relevance to this paper so prefer not to include them.

(ii)     When we did our summary statistics, we computed the geometric mean for soil pH in accordance with the IUPAC

| | |
|---|---|
| (ii) L218-220: I wonder how the mean of the soil pH was calculated. This is because it will be incorrect to just calculate the arithmetic mean of a phenomenon (like pH) that is on a log scale. | recommendations 1994 (Currie, L. A., & Svehla, G. (1994). Nomenclature for the presentation of results of chemical analysis (IUPAC Recommendations 1994). Pure and Applied Chemistry, 66(3), 595–608. https://doi.org/10.1351/pac199466030595). To make this clear we have edited L216 to:

We undertook exploratory analysis of soil pH and Se$_{grain}$ concentration using QQ plots, histograms, and summary statistics (e.g. used geometric mean for soil pH and arithmetic mean for Se$_{grain}$) to check whether there was need for transformation of the variables for the assumption of normality. |
| (iii) L228-230: Any specific reasons for these minimum and maximum grid spacings? | **(iii)** The grid spacings were considered because the span the axis from finer grid to a coarser grid to fully illustrate the different prediction intervals that can be achieved by sampling effort. |
| (iv) L231: "We considered different prediction for each variable, but the prediction interval was fixed, depending only on grid spacing. The three predictions of soil pH were 4.8, 5.5 and 6.0 and those of Segrain were 20, 55 and 90 µg kg−1."à I can understand the need to keep the same prediction intervals, but considering that the soil pH as a soil property and Segrain as a plant property will be subjected to different dynamics of spatial change, was there a way to account for this in the predictions? | (iv) From our previous study Gashu et al. 2020 we showed that it is possible to examine spatial variation of soil and grain properties, sampled on an appropriate joint sampling design, by using model-based statistical analysis. The empirical best linear unbiased prediction (E-BLUP) has the allowance to add collocated and non-collected data and has an associated prediction error distribution that allows account for the differences in the predictions. |

| (v) | L233: What kind of chart? Is it Figure 1? If so, then please state it. | (v) | This chart refers to Figure 1. To make this clear, we will edit the text on L233 so that the readers can be sign posted to Figures 1, S7 and S8. |

The predictions of soil pH and $Se_{grain}$ concentration were presented to the participants in a chart (see Figure 1, S7 and S8).

| (vi) | L234: "From the chart, we asked the participants to select the grid spacing that gives the widest prediction interval that would be acceptable if the mapped predictions were to be used to make decisions about soil management or interventions to address human Se deficiency." à I find difficulty in embracing the premise upon which this question is constructed. Initially, my understanding was that the inquiries were primarily concerned with the planning of a geostatistical survey. Therefore, it confounds me as to why participants are questioned about employing the maps as a foundation for decision-making. In a theoretically optimal scenario, what would constitute the best choice of a prediction interval for such a decision? | (vi) | The whole point of a survey is that it produces predictions, and the basic premise of our study, which we hope will be clear in the revised paper. Also, that prediction quality responds to survey effort. The geostatistical methods can capture the measures of the quality of spatial information explicitly. We have proposed changes in the introduction which makes it clear that the width of the prediction interval depends on the conditional prediction distribution and so on grid spacing. |

| (vii) | L240-245: If a conditional probability of 1 indicates that the prediction is a equivalent to the overall mean of the dataset, does it suggest that the conditional mean of <1 is an indication of underestimation or over estimation of the true value at the given location? Also, I have the same issue with the question posed here as that posed in L234 above. L245-263: My concerns mirror those I previously expressed in section 234. | (vii) | The probability goes to zero or to one because the prediction goes to the mean which either indicates the intervention or not, depending on the threshold and the mean value. |

(viii)   Participants are queried about interventions, but their responses are then utilized as a foundation for planning a geospatial survey. This connection appears incongruent, and it might be worth clarifying how the answers to these questions directly inform the planning process.

(ix)   Section 3.2.5: I'm grappling with a particular aspect of the offset correlation method, namely its use as a measure of similarity between two grid spaces. For this measure to function meaningfully in decision-making, one grid space must be taken as a reference, representing the closest approximation to reality. Then, higher correlation with this given reference space would indicate an optimal choice among the others. However, in the method's current presentation to participants, an issue arises. Specifically, there's a risk of bias propagation; grid spaces that are closer together are likely to show higher correlation compared to those farther apart. Similarly, coarser grid spaces might exhibit greater correlation across the board. These biases can distort the method's effectiveness. How did the authors address this potential source of error?

(x)   Figure 4: Please check, the caption mentions Segrain, but the figure indicates soil pH. Also, it would be meaningful for the reader to know the grid space of map1 and map2 that is giving the correlation value of 4. As I have indicated in my comment

(viii)   As noted above, it is fundamental for these approaches to survey design that the information is used for a purpose. We have made this clear, in the revised materials and method section.

(ix)   We hope that our proposed revisions (above), ensure the offset correlation concept is well understood by a broader audience. Making the grid spacing coarser always increases the offset correlation.  The point is that we posit two maps based on the same grid spacing but offset by the maximum possible distance in each axis (half the grid spacing). Neither map is expected to be closer to reality than the other, the question is how consistent they are. The best analogy would be when a lab does triplicate analyses on some soil samples.  We do not say "one of those three analyses must represent reality and we compare the other two with them". Rather, we say, if our method is good enough then the three measurements should be consistent with each other. We wish to add the following at L170:

Neither of the posited pair of maps, based on offset grids, is to be regarded as closer to reality than the other, the question is how consistent they are.

(x)   The caption has been edited to reflect the referee's suggestions.

Figure 4. The pair of hypothetical maps of pH value and corresponding scatterplot for offset correlation 0.4.

| | |
|---|---|
| above, it will be useful to know which of these two is closer to reality. | As noted above in (ix), there is no reason to believe that one map is closer to reality than the other. The question is how consistent are they? |
| **3.3 Data analysis**

Section 3.3.1: "The expected number of responses under the null hypothesis, ei,j in a cell [i, j], is a product of row (ni) and column (nj) totals dived by the total number of responses (N), and this the null hypothesis of the contingency table which is equivalent to an additive log-linear model of the table" à What is intended by this sentence? Please consider revising it to be more comprehensible for readers who may not be statisticians. For example, instead of stating 'Contingency tables allowed us to test the null hypothesis of random association of responses with the different factors in the columns,' it would be more helpful to specify what the null hypothesis was in relation to the different responses. This clarification would illuminate the process and make the statement more approachable for a broader audience. | Thank you for this suggestion; we propose to edit the statement on L275 to

The contingency table is analysed on the basis of a null hypothesis that the distribution of observations between responses (e.g. selected grid spacing) is independent of the factor in the column (e.g. professional group). If evidence is provided to reject the null hypothesis, then this would indicate that how a respondent interprets the information presented to select a grid spacing depends on their professional group |
| Table 2 appears to neither enhance the flow of the paper nor contribute to its content. Consider relocating it to the appendix, where it can be accessed if needed without interrupting the main narrative of the paper. | Table 2 will be moved to the Appendix. |
| Section 3.3.2: "However, in our analysis we reversed the order by assigning a score of 4 for the most preferred method and 1 for the least." à Why was it necessary to do this? Why wasn't it possible to also offer to the participants the same way you analysed the data? Perhaps, you could have also tested if the sequence of the choices offered would have had an effect on the decision. | We did this to assign a score of 4 to the most preferred method, and 1 for the least to the most ranked method. The mean rank is computed from the product of the rank and score dived the number of participants.

However, it makes no difference reversing the order or not. The respondents ranked the methods in order of their preference. To make it clearer, we propose to edit the statement on L318: |

| | However, to calculate the mean rank, r, for each method for all the respondents, we assigned a score of 4 for the most preferred method and 1 for the least. |
|---|---|
| **4 Results**

Section 4.1 test methods can be removed as there is no text under this section | Sub-sections 4.1.1 to 4.1.4 all are under the section 4.1 which presents the results from test methods. Then 4.2 presents results from assessment of the methods. Therefore, this heading is necessary to make this distinction. |
| Section 4.1.1 presents a discrepancy in the order of the methods, with the 'offset correlation' appearing last in the methods section but first in the results. To enhance clarity and consistency, I recommend aligning the order of appearance in both sections. | The suggestion will be included in the revised paper. |
| L338-340: From what I understand so far about offset correlation, the correlation value is combination pair of two grid spaces, so what does it mean here that the grid spacing for soil pH is 25 km and that for Se$_{grain}$ is 12.5 km? What is the other pair in this correlation combination? Also, from figure 5, it can be seen that while most people indicated 0.7 correlation value, there were still a substantial number of people that selected the full range of the correlation values. Given the low number of participants (n) it will be useful to not only report on the most but also critically consider the other correlations. I think this is one of the major flaws of this study. | The summary given here does not reflect how the offset correlation is defined. The offset correlation is dependent on the variogram model of the property- we had two variograms one for soil pH and the other for Se$_{grain}$. so there is no reason to expect that the offset correlation will be the same for two different variables at a given grid spacing. We will revise the paper to improve our explanation (see comments above). In the revised paper we will comment on the range of values for the offset correlation. |
| Section 4.1.2: L345, do you mean there were no differences considering the p-value you reported? While I agree that the reported p-values suggest that the null hypothesis of uniformity in response cannot be rejected, it can be see from Figure 6 that the percentage of people that selected the grid spacing of 100 km (< 5 %) were substantially lower compared to the rest of the population, so what accounts for it? | While there is a fluctuation at 100 km the analysis tells us that it is potentially misleading to look for an explanation as the overall result is quite compatible with a random distribution. |
| **5 Discussion**

L383-385: "In this study, we presented to groups of stakeholders, four methods (offset correlation, prediction intervals, conditional probabilities and implicit loss functions) that can be used to support decisions on | We will change stakeholders to "information user" and this change will be made on L383-385 and the rest of the manuscript. |

| | |
|---|---|
| sampling grid spacing for a survey of soil pH and Segrain."à I don't think you can regard your participants as stakeholders in this case. It still remains to be answered what is at stake for them. | |
| L385-390: "Offset correlation was ranked first as the method the stakeholders found easy to interpret (see Figure 9), and over 70% of the stakeholders specified a correlation of 0.7 or more as a criteria for adequate sampling intensity" à I am unsure where this 70 % is coming from because from Figure 5, it is only 30% that chose that 0.7 correlation. Since, the 0.7 value was chosen as part of an ordered categorical set of variables (from 0.4 to 0.9), it is inconsistent to draw a conclusion like "0.7 or more". As I have already indicated in an earlier comment in the results, it is equally important to know why people chose 0.4 or 0.9 as their best choice of correlation coefficient for intervention. | We do not agree that this is an inconsistent interpretation. We clearly state that "**correlation of 0.7 or more.**" This means >30% for 0.7 plus 28% for 0.8 and ~15% for 0.9 which is over 70%. Furthermore, a respondent who thinks that an offset correlation of 0.7 is necessary would regard a design for which the OC was 0.8 as acceptable with respect to quality, but one for which it was 0.6 as not, so there is an asymmetry, and we can state that 70% of respondents thought that an offset correlation of 0.7 or more was acceptable. We will edit the sentence L385 to make it clear:

Offset correlation was ranked first as the method the stakeholders found easy to interpret (see Figure 9), and most respondents (30%) selected an offset correlation of 0.7, and slightly fewer selected 0.8 so over half of respondents are accommodated within this range of values. |
| "During the feedback session, stakeholders highlighted that they were more familiar with the concept of correlation, with a closed interval of [0,1]. This explains why there more consistent responses under this method." à This here is another major flaw in the whole study. Was it the stakeholders that selected 0.7 who made this declaration or was is it also the same for those who chose 0.4 and 0.9, because if the concept of correlation is familiar, then you would strive for a stronger correlation of 0.9 and not 0.7. Also, it is wrong to indicate that correlation has a close interval of 0 to 1, because the interval of correlation is -1 to 1. | We propose to make the following change on L413:

This is likely to explain the consistency of the results for this criterion, with over half the respondents selection 0.7 or 0.8 as a minimum acceptable correlation.

The question to the respondents was not "what correlation indicates the best design" but rather, what is the minimum acceptable correlation. That is very different.

Also, to mention the offset correlation ranges from zero (when the maps produced from the two grids are independent of each other (at a coarse spacing) and approach 1 as the grid becomes finer and the two maps become increasingly similar. We will add the following text on L170: |

| | The offset correlation is bounded [0,1], and ranges from zero (when the maps produced from the two grids are independent of each other (at a coarse spacing) and approach 1 as the grid becomes finer and the two maps become increasingly similar. |
|---|---|
| L390-393: "Our results are consistent with findings of Hsee (1998), that relative measures of some uncertain quantity (Hsee gives an example of the size of a food serving relative to its container) are more readily evaluated than absolute measures (the size of serving). An easy-to-evaluate attribute, such as the bounded correlation of [0,1], has a greater impact on a person's judgement of utility. Hsee (1998) describe this as the "relation-to-reference" attribute. It is therefore, not surprising that the offset correlation is highly-ranked." à As I have already explained above, correlation is not bounded between 0 and 1, and the fact that authors' failed to grasp this clearly indicates that it is not a simple "easy-to-evaluate" attribute. It will be helpful for readers if the greater impact of an easy-to-evaluate attribute on judgement of utility is explained, given that this seem to be one of the main conclusions from this study. It will also be useful if Hsee(1998) "relation-to-refence" attribute can be explained as to how it relates to this study. | The offset correlation is bounded between 0 and 1, *pace* the reviewer. This is not difficult, many correlation measures used in statistics are non-negative such as intra-class correlations or heritabilities. We will revise our explanation to make this explicit. |
| L394-395: "The offset correlation will be more useful for stakeholders who are not able to express their quality requirement for information in terms of quantities such as kriging variance." à Was this statement also derived from the feedback session? In which way will it be more useful? | We will make the following change in L394:

The offset correlation seems to be a criterion which respondents are more likely to find comprehensible, and so a basis for selecting the sample density for a geostatistical survey, than alternatives such as kriging variance. |
| L395-399: "Furthermore, it is an intuitively meaningful measure of uncertainty, it recognises that spatial variation means that maps interpolated from offset grids will differ but that the more robust the sampling strategy the more consistent they will be. There is a paradox | We will make the following change from L395 to reflect this.

Furthermore, it appears to be a measure of uncertainty which participants in the study found comprehensible, and so were able to use to select a grid |

| | |
|---|---|
| here, however, in that the previous study Chagumaira et al. (2021) showed that interpretation of survey outputs in terms of uncertainty was easiest for stakeholders with measures related directly to a decision made with the information. The offset correlation is a general measure, and the absolute magnitude of uncertainty." à I wonder how offset correlation is an intuitive measure of uncertainty, can you please explain. And can you also explain the paradox you mention? | sample spacing. It recognises that spatial variation means that maps interpolated from offset grids will differ but that the more robust the sampling strategy the more consistent they will be. However, Chagumaira et al. (2021) found that measures of uncertainty related to a specific management threshold of the mapped variable were preferred by participants for the interpretation of uncertain spatial information to general quality measures without a specific management or policy implication. In this case, in contrast, the preferred criterion, the offset correlation, is a general measure of map quality, which is not directly linked to specific interpretation. |
| L403-411: Interesting explanation. I wonder if author's don't find it strange that the same people (stakeholders/participants) that could understand the bounded attribute [0,1] of the offset correlation cannot seem to understand a similar attribute of the conditional probabilities simply because it is "probabilities"? | We do not find it strange. Offset correlation [0,1] tells us that the maps made from offset grids are either completely independent of each other (0) or identical (1). In contrast the probability is (i) conditional on data and (ii) is a joint probability so it measures the overall probability of making a particular error in interpretation of the information: failure to recommend an intervention, resulting from (a) the uncertainty of the prediction and (b) the overall probability of the corresponding intervention being required. |
| L411-418: As I have already indicated in the results section the response on the grid spacing of 100 km is markedly lower than the rest, so I expected some explanation as to why this is so. The explanation given here is too superficial and inadequate to explain such a complex decision-making process. | Our analysis shows that there was no evidence to reject the null hypothesis that the responses are uniformly distributed see Table 4. The result is quite compatible with a random distribution. It would be potentially misleading to give an explanation to as why there are fewer responses on 100km, yet the evidence suggests these responses are uniformly distributed. |
| General comment: Based on the issues I've highlighted throughout my review, it's apparent that the remainder of the discussion and conclusion sections also warrant similar concerns. I strongly recommend a comprehensive revision of the manuscript to explicitly delineate its unique contributions to this most important field of science communication and particularly on communicating uncertainty. | We thank the referee for thorough review of our work, and we believe we have addressed all the concerns raised. |

---

## Author Response (AR2)

**Author Responses to Referee Comments:**

We would like to thank the referees for their thorough review of our manuscript. We have revised the manuscript based on their suggestions and comments. We reply to each of the comments below. Our changes in the paper are in blue below, and the line numbers and sections refer to the revised manuscript:

**Referee Comments 1 (RC1):**

| Referee Comment | Author Comment |
|---|---|
| 1. I appreciate the effort that has gone into improving the contents of this manuscript, especially regarding the extensive structural and content changes based in part on my concerns. Unfortunately, while the basis of the study is interesting, most of my concerns remain unresolved or only partially addressed in the author response. In my previous evaluation, I suggested focusing on the human decision process in planning spatially explicit surveys, which is purportedly the focus of this paper. This required a thorough restructuring to understand the perspective of respondents, the scenarios they were presented with, and why only 26 people were selected or participated. However, the revised manuscript only mentions that respondents "self-identified" as "x-y-z" professionals, without independent verification. There was no attempt to explain the statistical power of the 26 people, its limitations, and how (un)representative the 26 people are for their representative professions that they seem to be speaking for. This is especially concerning given the small sample size and the overrepresentation of agronomists (18 out of 26). Considering that these respondents are practitioners in their various fields, it would have been useful to know their background information such as age, sex, and years of experience, and most importantly if they have any involvement in planning geospatial surveys amongst others. This missing context is crucial for | We appreciate your suggestions regarding the participant composition and representativeness. In this study, we aimed to capture perspectives from professionals with direct, self-reported experience relevant to geospatial survey planning, particularly focusing on individuals actively involved in agricultural and nutrition fields. The intention was not to generalize findings beyond this group, but rather to explore practical insights on decision-making processes from those currently involved in survey design.

Regarding background and sample size (see responses 9 and 8 below)

We appreciate this thorough reading of our paper. However, we do not agree that the primary focus is on the decision, rather it is on the means to communicate the information required for that decision. We agree that the actual decision process is complex and specific (and have addressed this elsewhere, e.g. Chagumaira et al 2022). We have tried to clarify this at various points in the abstract (L1—L21), introduction (L91—L100) and discussion (L427—L450), and have also made an edit to the title of the paper.

We have addressed the issue of the size of this experiment in our response to RC3 (see responses 8 and 9 below). On the issue of `self-identification' this was not to define a population of stakeholders in terms of expertise. The population was defined as professionals in the listed fields who were engaged with Universities in the Gates-funded GeoNutrition |

| readers to understand why respondents would prefer one method over another. | project and the allied UK Research Council-funded Zim-GRTA project. They were identified by the country lead on the project. By virtue of contributing to this project (e.g. as extension officers in national extension services or national public health researchers or nutritionists) the participants were in the target population. The self-identification was simply to show how the different specialisms were distributed- we clarify this on L119—L123. We regarded the mathematical background and experience as essential additional information, but could not justify collecting personal information, such as gender, to the Research Ethics Committee. It must be emphasized again that the objective was not to elicit a sample density, but rather to examine the comprehensibility and usability of alternative measures of how uncertainty and sample effort are related. |
|---|---|
| 2. It is particularly worrying that authors consider only studies involving "indistinguishable pills" could be blinded or randomized in an experiment, and that randomization or independence of samples is irrelevant in sociopsychological studies such as this one. This kind of social experiment has strong theoretical basis in economics of "choice experiments", and I expected the authors to ground their work in such an exemplary well-established theory. Indeed, this is quite surprising that such a theoretical foundation is missing for a study that is so heavy on frequentist statistics such as this one. From my assessment, most changes in the revised manuscript are largely editorial and do not address the core content I previously suggested. | Our comment on " indistinguishable pills " was specifically in response to the original suggestion that double blinding was an appropriate approach. It is surely self-evident that an experiment in which a researcher explains a set of methods to communicate information to a participant, who then applies these methods, and records their impression, cannot be "double-blinded". The researcher cannot explain a method without knowing what it is! Even the most experienced of agronomists could not apply and rank a set of methods, without knowing what they are.

Regarding randomization, we gave the explanations of the methods in an order which we believed would allow participants to build their understanding of geostatistical predictions and their uncertainty, starting with the simplest concepts, then introducing the more complex. Randomizing this order would only cause confusion. We accept that randomizing the order of tasks would be ideal. However, we felt that it was appropriate to let the participants approach the task of applying the different methods as they found comfortable. They could approach the tasks in the order they preferred. For an exercise necessarily undertaken online, we felt that this was the most appropriate way to collect useful |

| | responses. We discuss this as a matter for further work (see L426—L440). |
|---|---|
| 3. Moreover, it seems arbitrary to treat respondents from four different countries (Ethiopia, Malawi, Zambia, and the UK) on two different continents (Europe and Africa) as having comparable experiences and backgrounds. One wonders what soil scientists in the UK have to do with a study designed to understand the planning of a geospatial survey using the context of a province in Malawi. Under what circumstance are soil scientist in the UK grouped together with agronomist and public health professionals from four African countries? It is insufficient to claim that respondents are motivated by some MND project goals, as if they are a homogenous group of people. The paper inaccurately assumes that by eliciting information from these 26 self-styled professionals provides broader insights into the planning process of geospatial surveys. This is clearly not the case. Not only are the thoroughly explained geostatistical methods and the survey approach lacking novelty, but the study also wrongly assumes a unidimensional simplification of multidimensional complex human decision-making process. | The participants were all engaged in a common project, bringing their expertise and experience to bear. We agree that there could be differences in the *decision process* regarding an intervention in UK and African contexts in so far as the losses arising from decisions, which are suboptimal because of the uncertainty in the information on which they are based, would differ. However, our focus is not on the decision process as a whole, but on the accessibility of the different forms of information which we have presented see introduction (L91—L100), methods (L119—L123) and discussion (L426—L450). We think that this will reflect the focus of education, training and experience, but see no reason to expect a systematic difference between UK and Africa-based professionals.

The reviewer makes some rather sweeping statements about the novelty of the statistical measures of uncertainty that we trialled. The kriging variance was proposed as a statistic for sample design in the early 1980s, there are relatively few hard examples of its application in the way proposed. The related confidence interval has also been proposed, but the particular way we have expressed it for interpretation is new and based on our previous experience (Chagumaira et al 2021, https://gc.copernicus.org/articles/4/245/2021/ ) using prediction intervals as measures for the assessment of uncertainty in spatial information. The offset correlation was first proposed by one of us (RML), but this is the first attempt to use it with a stakeholder group. The joint probability we use in this study has not been proposed for this purpose and is the first attempt to use a generalized uncertainty measure which accounts for the location parameter of the variable as well as its spatial dependence (see L95—L100). For this reason, we think that the comparative assessment of how these different methods is received and applied is a useful contribution. With regards to location of respondents please see response 1 (above). |

| 4. What is most disturbing is the superficial confirmatory arguments throughout the paper. For instance, we are told that respondents chose Off-set correlation because, in an unrelated study by Hsee (1998), people prefer bounded attributes over absolute ones. Yet, this same explanation does not apply to the joint (conditional) probabilities with similar boundaries, due to their "probabilities". Such superficial empirical studies directly contradict the rigorous 'Popperian' falsification advocated in modern scientific inquiry. In my previous review, I advocated looking into the reason behind one person (representing 5% of the 26 people) who selected 100 km grid space as the optimal, and the authors' response is "… the analysis tells us that it is potentially misleading to find explanation", as if the "analysis" is absolute and final. This rather bizarre answer is another indication of how the study made no attempt to find alternate explanations to their confirmatory responses. | Our aim was to find out to what extent are these methods usable by stakeholders from different discipline, and to highlight general trends in survey planning preferences among respondents, while recognizing individual variability.

The reference to Hsee (1998) was intended as a supplementary point rather than a definitive explanation. It started from the observation that the offset correlation was generally preferred. The offset correlation takes values in [0,1] (it cannot be negative) which corresponds to a range from "zero information" to "perfect information". This clearly relates to Hsee's conclusion. The conditional probability does not have such a simple interpretation. We noted (see line 406 of the original paper) that is also bounded on the interval [0,1], which might explain why it was ranked highly even though it was generally misinterpreted. We think that the fact this "relation to reference" effect appears to explain the high ranking of offset correlation and conditional probability, even though the latter was clearly widely misunderstood, is an important finding, related to our key objective. We have added the following sentence from L374:

A method might be regarded as easy to interpret, because of its form, even when it is not (in this case a large value of the probability indicated that there was no spatial information in the map to make its predictions better than the overall mean).

The way in which the conditional probability is specified depends on the problem for which the spatial information is used (a threshold), and whether a large or small value is preferable depends on its exact formulation. This was explained to participants, but clearly it was less accessible than the offset correlation. We make this point at L95—L100 in the revised paper. |

| | We do not think that the question of falsifiability is relevant here. We are not testing hypotheses about decision making. We are making a practical evaluation of the extent to which each method succeeds in communicating how uncertainty depends on the method used. |
|---|---|
| 5. After reading the entire paper, I still feel that the four proposed methods are not directly and necessarily comparable as presented in this study. The underlining explanations of the various geostatistical methods do not indicate the need to choose one over the other. Thus, study participants may be subjected to a hypothetical situation that is neither necessary nor realistic. The revised manuscript explains: (a) Prediction interval is based on the kriging variance (which is estimated from the empirical variance taken from an earlier survey data?); (b) Joint (conditional) probability assumes a location requires some intervention (what sort of intervention?) based on the kriging variance, indicating that a prediction does not correspond to a particular threshold set a priori (obviously more information is needed to understand this); (c) Implicit loss function, which is not based on the kriging variance but on a hypothetical loss (but it is unclear whether this loss is economic or information loss or both) for making a spatial decision that is correct or an error (where it is undefined what constitutes correct and erroneous decision); (d) Offset correlation is based on the consistency of spatial information at two hypothetical grid spacing, not on the kriging variance. Yet, it remains unexplained why these four methods, requiring different data inputs and different underlying assumptions, need to be compared as this study did. None of the data and resource constraints used to build scenarios for participants make it reasonable to choose one method over the other. The methods may be useful under different conditions and applicable situations, which may not necessarily overlap requiring | The methods are not all directly comparable, and this is clear. However, they are all derived from a common statistical model of the variable, and so are mutually consistent.

 Our view is that all the methods could be used in the proposed scenario but provide different information. The information provided in each case was explained to the participants. What we learn from their responses is how far they regard themselves as being informed by the information about how uncertainty and sample effort are related. We clarify this point in the revised paper at L95—L100. |

| users to use one instead of the other, for planning a geostatistical survey. Given the lack of basic information on the comparability of the methods, the missing key information of the respondents, and the lack of theoretical grounding, along with the fact that almost all my concerns are adequately unresolved, I cannot endorse the publication of this manuscript. Additionally, the paper needs a careful editing for grammar and sentence clarity as many sentences were impossible to parse. | |
| --- | --- |

**Reply on RC3**

| Referee Comment | Author Comment |
|---|---|
| 6. This paper had a number of strengths, including a well-written and robust introduction section and literature review. It was a pleasure to read an introduction that so eloquently explained the wider research project, implications of that research, and why you are trying to find the most effective means of communicating uncertain data. Additionally, you have highlighted an area of study not yet discussed in the scientific literature. I have a few general recommendations in addition to some specific edits: | Thank you for the opportunity to revise our manuscript. We would like to thank the referee for their time and for the constructive comments they have provided |
| 7. This paper is very long, I recommend consolidating information and monitoring the paper for conciseness. I realise it is hard to balance thoroughly explaining the background and considering length, but as a reader, I became lost and fatigued trying to understand all the communication methods you tested. It seems this could be shortened considerably. Perhaps one way of doing so is to show the examples in section 1.3 with a very brief explanation of what they mean, for example, a kriging method to determine probabilities of different pH. | We thank the referee for their feedback. We have revised the paper and made some statements more concise. |
| 8. There is little to no discussion about the the impact of sample size on your results. There are very few people involved in your study, which impacts the robustness of your results. It's likely that there are statistically significant findings in your method, but there are too few people in the sample to suss that out. I recommend extending this study to additional people, including entire departments from the universities involved in the study, governmental bodies involved in the study, etc. For such a small sample size, the focus of the results should be on more qualitative data, for example the feedback sessions rather than statistical analysis of results. I'm particularly concerned about the results from the tests you ran comparing the groups divided by | We thank the referee for this comment, and we make the following change from L413:

 All the information users recruited in this study were employed in public sector institutes (e.g., universities, civil organisations, research, and extension) and had experience in their respective fields in an SSA setting. We had no basis for a power analysis to identify a sample size for this activity. Given the exploratory nature of this research, our primary aim was to capture insights from as many relevant participants as possible within each institution. As a result, our major consideration was recruiting individuals willing to participate and with experience in their respective institutions. We therefore attempted to recruit the entire set of suitable |

| | |
|---|---|
| specialism - 8 people in one group doesn't make a robust statistical analysis in a survey. | respondents in each country. We recognize that the small sample size limits the generalizability of statistical findings. While this study provides insights into participant perspectives by specialism, the lack of demographic information—such as gender, age, location, and years of experience—limits the depth of analysis. These characteristics may impact responses; for example, different age groups or experience levels might prioritize certain issues differently. Future studies should consider including these demographic details to explore how such factors influence perspectives, thus enhancing the robustness of the findings and allowing for subgroup analysis. For this reason, we have interpreted results cautiously and have also incorporated qualitative insights from participants to provide a richer context for understanding these early findings. Moving forward, we plan to include an initial power analysis and possibly extend the study through broader collaborations to enhance robustness. |
| 9. The only information you give on participants is the number of participants in each specialism. Please include information such as gender, age, location, years of experience in their fields, etc. Add to the discussion how these characteristics may impact your survey results | We have added information on the composition of the participants in the appendix and the following text from L285:

There was reasonably even spread in terms of the location of our participants, see Figure B1 (Appendix B). About 54% of the participants were constantly using statistics/mathematics within their job role. Only a few participants were educated to the level of certificate/diploma (8%).

We also have edited the following in the discussion from L419:

While this study provides insights into participant perspectives by specialism, the lack of demographic information—such as gender, age, location, and years of experience—limits the depth of analysis. These characteristics may impact responses; for example, different age groups or experience levels might prioritize certain issues differently. Future studies should consider including these demographic details to explore how such factors influence perspectives, thus enhancing the robustness of the |

| | findings and allowing for subgroup analysis. For this reason, we have interpreted results cautiously and have also incorporated qualitative insights from participants to provide a richer context for understanding these early findings. Moving forward, we plan to include an initial power analysis and possibly extend the study through broader collaborations to enhance robustness. |
|---|---|
| Line 176: I'd like a sentence or two on what Se is and what a lack of Se in the diet does to a person. You mention the average requirement of Se in adult women, but that leads the uninitiated reader unclear on why they should care about Se concentration | We thank the reviewer for pointing this out, and we have added the following text from L184:

Selenium is an essential micronutrient with critical roles in human health, and lack of it can cause thyroid disfunction, and suppressed immune response (Fairweather-Tait et al 2011). |
| Lines 381-384: From my memory, this is the first time the feedback session was mentioned. This should be discussed at the beginning of the methodology section and more focus should be given to this part of the study as qualitative feedback is more robust with a small sample size. | We thank the referee noticing this and we have added the following from L160:

We had a feedback session to allow the participants to seek clarification on the presented methods. |
| Lines 398-401: The statement of "it is not clear how to select an appropriate effect size" is unwarranted. There are a lot of peer reviewed articles involving surveys and expert elicitation, | Thank you for this observation and we have addressed this comment, see response 8 |

---

## Author Response (AR3)

**Author Responses to Editor Comments**

We would like to thank the Editor for the thorough review of our manuscript and suggestions made. We have revised the manuscript based on the Editor's suggestions and comments. We reply to each of the comments below. Our changes in the paper are in blue below, and the line numbers and sections refer to the revised manuscript:

| Editor Comment | Author Comment |
|---|---|
| While I do share some of the concerns with reviewer 1 and believe this need be addressed to make further progress on this front in the future, GC takes a more "pragmatic" approach to sharing discovery (not strictly requiring to dive deep into the underlying theory to explain an signal/observation as long as robust quantitative or qualitative observations are being made). I do believe that this exploratory study is of value provided honest and correct framing (which authors have already improved during revision). However, the authors have not fully addressed comments by reviewer 3 regarding length and conciseness. While the authors have pointed out in their response that they worked on the conciseness of statements, the tracked-changes document shows more additions than deletions in the document, making the manuscript even longer. The impact of this on the use(fulness) of the manuscript is not to be underestimated, and I agree with the reviewer that parts of the manuscript could be "shortened considerably". I encourage the authors to go through the manuscript, consider carefully what information is needed, identify information that may be non-essential to the manuscript's message, and shorten it accordingly. Examples of where text could be shortened include (but are not restricted to): | We thank the referee and editor for their feedback about the length of the manuscript. We have thorough revised the manuscript to reduce the length. We outline below how we have done so, and our deletions & revisions are shown clearly in the track changed version of our manuscript.

We also have moved the Appendix to the Supplement to shorten the manuscript as well. |

| | |
|---|---|
| 1. Section 1.1 could be shortened to a third if condensed to information that is essential to communicating the work. | 1. We have shortened Section 1.1 (L25 to L44) and 1.2 (L45 to L54). |
| 2. Much of the information in the 1st paragraph of "Materials and methods" may not be needed to understand this study. Simultaneously, other information readers may initially be looking for, such as sample size, should be displayed more prominently. By being more concise, the section could probably be reduced to 1/2 or 2/3 of its current length.

I appreciate that this may be a challenge but keeping the text to-the-point and concise will enhance clarity, accessibility and ultimately the usefulness of the study to the community. A colleague who was not directly involved in the write up may be able to help identify essential and non-essential information. | 2. We have re-structured and shortened the first paragraphs on the Materials and Methods section (L94 to L115). We have put information about sample size on L108 to make it more prominent for the readers. |
| Other points:

1. Make sure you point out the methods are not entirely comparable as they communicate different aspects (see reviewer 1 comment). In a carefully set up (future) experiment, the number of free parameters should be minimised. | We thank the editor for the following suggestions and have addressed them:

1. We have made this clear in the abstract see L13 to L15, also on L87 to L90 in the Introduction. |
| 2. The authors end the abstract with "The offset correlation will likely be more useful [...]". Given the exploratory nature of the study and the sample size problems highlighted by the reviewers, I recommend adding a note about the need of a more in-depth study (with larger sample size, fewer free parameters, etc.) to really say this with confidence. | 2. We have added the following sentence at the end of the abstract, L22 to L23 "However, given the exploratory nature of this study, and the small sample size, there is need for more in-depth study with a larger sample size and fewer parameters to explore this further." |

---

## Author Response (AR4)

**Author Responses to Editor Comments**

We would like to thank the Editor for the thorough review of our manuscript and suggestions made. We have revised the manuscript based on the Editor's suggestions and comments. We reply to each of the comments below. Our changes in the paper are in blue below, and the line numbers and sections refer to the revised manuscript:

| Editor Comment | Author Comment |
|---|---|
| 1. The limited sample size, and implications of this, need to be clearly emphasised. Compulsory. | 1. We have emphasised the limited sample size see L23—L24,

However, the results should not be generalised due to the small sample size–there is need for a more in-depth study with a larger sample size to explore this further.

L405—L408

The events were planned prior to the lifting of all COVID-related restrictions on overseas travel from the UK and on larger gatherings in partner countries. Consequently, participant numbers were limited, and we recognize that these results should not be generalised due to the small sample size. To deepen our understanding, especially regarding the impact of professional grouping, a larger-scale elicitation is recommended. Conducting a face-to-face study would also be valuable to ensure participants fully grasp the probability concepts—particularly conditional probability—through interactive activities such as games and quizzes before formal evaluation. A practical takeaway is that more time is needed for participants to become familiar with the methods to improve the quality of the elicitation.

L421—L422
Given the small sample size in this study, there is need for a more in-depth study with a larger sample size to explore these findings further. |
| 2. Any remaining misleading confirmatory remarks i.e. "… as confirmed/supported also by study X/Y" when the studies aren't necessarily | 2. We have corrected misleading confirmatory remarks see L309—L305. |

| | |
|---|---|
| comparable need to be corrected as doing this misrepresentation and academic misconduct (see previous reviewer comments). Compulsory | |
| 3. This is a long paper for a rather limited result, so we strongly recommend shortening the paper dramatically, perhaps using figures to illustrate methods, to focus on the geoscience communication work. The consequence of not doing so is that very few people will read the paper. However, at this stage, we do not make this compulsory. | 3. We have shortened the methods (Section 2.2) and results (Section 3.1 to 3.2). |
| To help you, the Executive Editor team (includes applied statisticians) also made some detailed suggestions, all of which should be acted upon (i.e. compulsory).

4. Ethics should be in a separate statement. i.e. doesn't need to be on lines 111-115. | 4. We have moved the ethics statement to be a standalone, see L428—L431. |
| 5. Q4 doesn't make sense. Did they ask them to pick one? Or comment on all of them. Clarify. | 5. We have edited Q4 (Table 1) to make it clear. We were asking the participants to choose one of the three loss functions which would represent the loss incurred when a decision was made by using erroneous information. |
| 6. Section 2.2. I'd prefer far fewer words, and to see what the participants saw in all sub-sections, with detailed method in Supplementary Material | 6. We have shortened the text on Section 2.2 see L145—L158. |
| 7. Why is Q1 related to subsection 2.2.2, not 2.2.1. These need to be in order and clearly labelled as being linked. The linking becomes apparent when realising there are 5 sub-sections, but 4 questions. However, this highlights the need to | 7. We have revised this, and it is now in the same order throughout the manuscript. |

| | |
|---|---|
| try to write & structure the manuscript more clearly so that it's more readily readable. | |
| 8. Figure 1 should give me a sense of the decision and context being made – it doesn't do this at the moment e.g. perhaps a selection of scatter plots were shown to the participants. | 8. Thank you for the comment. We have revised Figure 1 to illustrate the offset correlation using simulated data, as we did not use actual study data to construct the pairs of maps shown to participants. In the revised figure, we have two hypothetical cases where we have offset correlation values of 0.4 (a) and 0.8 (b). In each case the illustrated subset of grid points is of the same dimensions, so the grid is denser in (b) than (a). In each case a hypothetical data set 1 (black grid points) and set 2 (grey points) is collected from grids of shared spacing but offset north-south and east-west by half the grid spacing. These map pairs and corresponding scatter plots were used during the task to help participants visually assess how much uncertainty is consistent with a particular offset correlation. We also have edited the caption to reflect the above. |
| 9. In Section 1.3, offset correlation is mentioned 4th. It should be in the same order that it's considered in the results and questions later. | 9. We have made the change, see L57—L64. |
| 10. In Figure 1, maps 1&2 should be visually related to a map including the original data and the location of the points of the two sampled grids, so that the process can be clearer. Things such as this (but not restricted to just this one example) may also help the authors shorten the text. | 10. Figure 1 illustrates the offset correlation using simulated data, and we cannot show actual data from two such grids because this was never actually done, it is a concept to illustrate the consistency that can be expected for a given grid spacing. |
| 11. Only statistically significant results should be interpreted. This may help you cut length and figures. | 11. We have described statistically significant results only see Sections 3.1.1 to 3.1.4. |

---

## Author Response (AR5)

**Author Responses to Editor Comments**

We would like to thank the Executive Editor for the thorough review of our manuscript and suggestions made. On behalf of the authors, I would like to express our gratitude to the thorough review process GC has, and our paper has improved significantly. It has been a long review but enjoyable and memorable experience.  We have revised the manuscript based on their suggestions and comments. We reply to each of the comments below. Our changes in the paper are in blue below, and the line numbers and sections refer to the revised manuscript:

| Comment | Author Comment |
|---|---|
| 1. Please add Q1 to figure caption for Fig. 5. This way the readers do not have to go back and forth between the figure and the manuscript text. Please do the same for Fig 6-8, and the relevant Tables where you mention any of the four questions. | We have added Q1 to figure caption for Fig 5 and have done the same for all figures and tables. |
| 2. Please shorten the title and put the focus on communicating uncertainty. One possibility would be something like "Communicating uncertainty to soil property map users" or "Communicating uncertainty and its dependence of sampling density to soil property map users" | We thank the editor this and we have shortened the title to "Communicating expected uncertainty of a geostatistical survey to support co-design with users of information" |
| 3. Line 8 in abstract - a verb is missing. | We have added the verb to Line 8 in the abstract "The first method, the offset correlation, is a measure of the consistency of kriging predictions made from data…" |
| 4. Consider adding more references for the statements made in lines 51-53. The authors state 'previous studies' but include one reference only. | We have made the suggested change on Lines 51—53 "Chagumaira et al. (2021) showed that non-statisticians often find the kriging variance difficult to interpret, and this is consistent with other findings on interpretation of variances by non-specialist (e.g. Konovalova and Pachur, 2021; Weber et al., 2004). It is unlikely that they would find it useful as a measure of the quality of survey output to balance against costs." |

| | |
|---|---|
| 5. Please review EGU policy on inclusivity in global research and where needed edit the manuscript accordingly (e.g., acknowledging all significant contributors including translators, in-country assistants, and external organizations that helped with data collection in Malawi and Ethiopia, etc.) - https://www.geoscience-communication.net/policies/inclusivity_in_global_research.html | We have reviewed the EGU policy on inclusivity in global research and confirm that all collaborators and significant contributors are acknowledged appropriately in the manuscript. This study formed part of my PhD research (https://eprints.nottingham.ac.uk/71710/), which was jointly conducted between the University of Nottingham, Lilongwe University of Agriculture and Natural Resources (LUANAR), and Rothamsted Research. Supervisors from LUANAR are included as co-authors on the paper to reflect their substantial contributions. |